# A column-like organization for ocular dominance in mouse visual cortex

**Pieter M. Goltstein** ⬥ ✉ **, David Laubender** ⬥ **, Tobias Bonhoeffer** ⬥ **& Mark Hübener** ⬥ ✉

The columnar organization of response properties is a fundamental feature of the mammalian visual cortex. However, columns have not been observed universally across all mammalian species. Here, we report the discovery of clusters of ipsilateral eye preferring neurons in layer 4 of the mouse primary visual cortex. These clusters extend into layer 2/3 and upper layer 5, forming a column-like pattern for ocular dominance. Our observation of such structures in this minute cortical area sets a new boundary condition for models explaining the emergence of functional organizations in the neocortex.

Cortical columns have traditionally been proposed to represent basic anatomical and functional units tessellating the mammalian neocortex[1,2]. Within the vertically oriented columns, neurons across cortical layers share functional properties, while across the cortical surface these properties typically change gradually, with occasional abrupt jumps, forming maps consisting of repetitive modules[3,4]. In the primary visual cortex (V1), this architecture gives rise to, for instance, the orientation preference map and ocular dominance columns[5,6]. However, while several maps have been found in mammals such as primates[1,7,8] and carnivorans (e.g., cats[9] and ferrets[10]), in rodents it is less clear to what extent the visual cortex is functionally organized.

Interestingly, the first electrophysiological studies in mouse V1 did observe a certain degree of functional clustering of orientation[11] and possibly also eye preference[12]. However, later work using two-photon calcium imaging did not find any obvious maps for these features[13–15], and it was a widely held belief that maps in visual cortex were largely absent in the mouse. More recently, some functional clustering at the micro-scale[16,17], ON/OFF domains[18], and potentially a global organization for orientation preference spanning multiple visual areas[19] were reported in mouse visual cortex. In addition, recent immediate early gene data suggest the presence of ocular dominance clusters in mice[20]. In rat binocular V1, a pattern of ipsilateral eye domains, most prominently identifiable in cortical layer 4, was reported using electrophysiology and immediate early gene labeling[21–23]. Thus, we asked whether clusters or column-like structures for ocular dominance in mice had been mostly overlooked, or whether they in fact do not exist, potentially because mouse binocular V1 is just too small a cortical area to harbor such an elaborate functional architecture.

## Results

In order to map visual cortex function, we used low-magnification two-photon calcium imaging in GCaMP6s transgenic mice[24] ($n = 9$), recording neuronal activity over an area of ~1 mm², covering mouse binocular as well as monocular V1 of the left hemisphere (Fig. 1a). Orientation tuning and ocular dominance of layer 4 neurons were assessed by presenting drifting gratings, moving in one of eight possible directions, to each eye independently (Fig. 1a, b). The resulting layer 4 volumetric recordings, spanning four imaging planes (spaced 20 μm apart), contained on average 5804 (±1042 s.d.) visually responsive (stimulus-tuned; see "Methods") neurons per mouse, with 4435 (±761 s.d.) neurons preferring the contralateral eye and 1369 (±517 s.d.) neurons preferring the ipsilateral eye. The ocular dominance index (ODI; ranging from −1 to +1, ipsi to contra; see "Methods") of visually responsive neurons was skewed to the contralateral eye (Fig. 1c, d) having a mean of 0.31 (±0.09 s.d.) across animals.

### Clusters of ipsilateral eye preferring cells in cortical layer 4

When inspecting HLS (hue, lightness, saturation) maps for ocular dominance (Fig. 1a), we found that some mice showed clear patches of cells with ipsilateral eye preference (Fig. 1c; Supplementary Fig. 1). In order to quantify these clusters, we used a local-density based clustering algorithm[25], allowing to identify patches of cells responding preferentially to the ipsilateral eye (ipsi-clusters; see "Methods"). We calculated the average ODI across all neurons (both ipsi- and contra-preferring) as a function of distance to the centers of the detected ipsi-clusters. If ipsi-clusters merely reflected an overall uneven spatial distribution of neurons, there should be a similar pattern for contralateral eye preferring cells, and the average ODI near ipsi-clusters should be

Max Planck Institute for Biological Intelligence, Martinsried, Germany. ✉e-mail: pieter.goltstein@bi.mpg.de; mark.huebener@bi.mpg.de

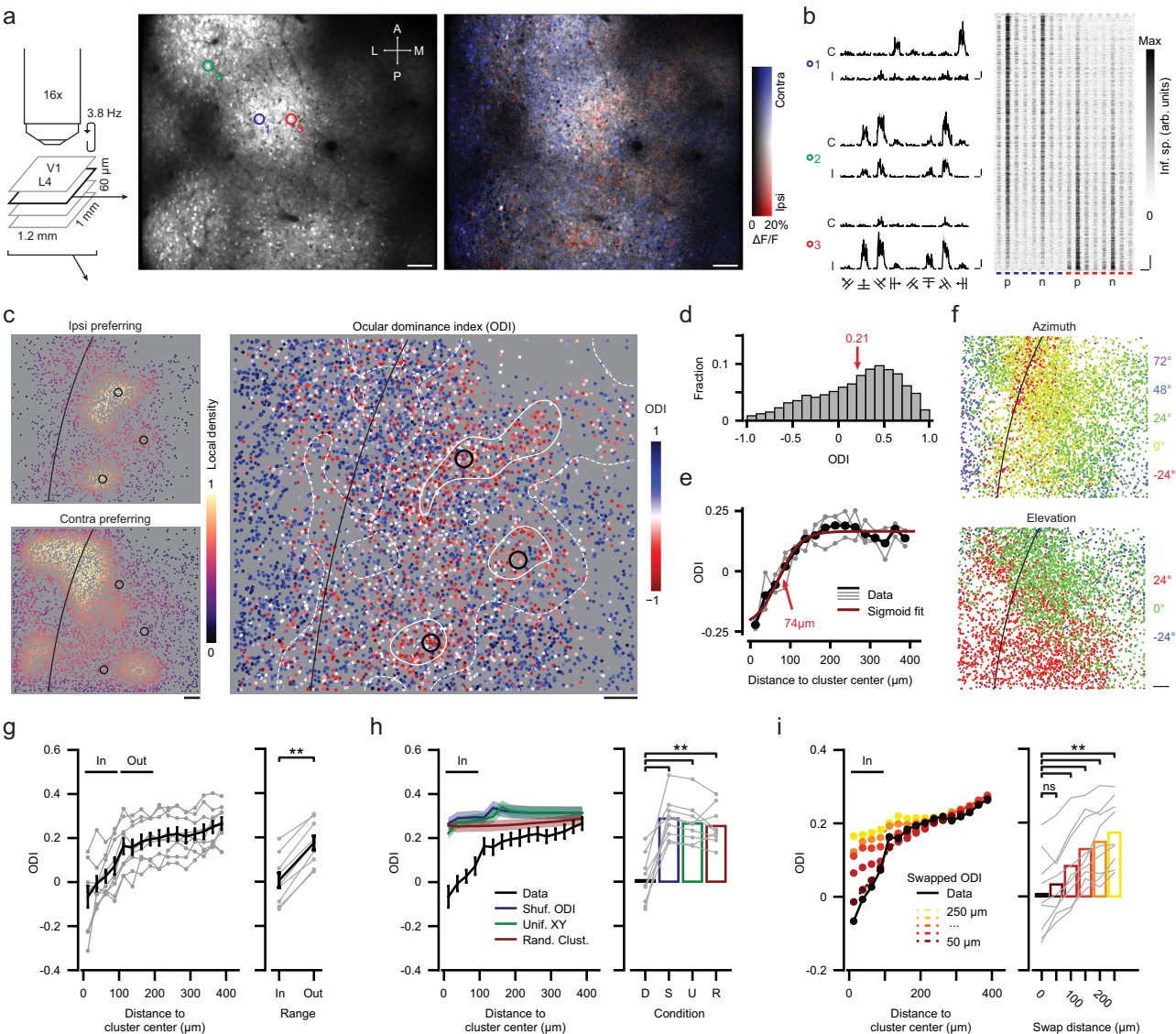

**Fig. 1 | Spatial clustering of ocular dominance in layer 4 of mouse visual cortex.**
**a** Multiplane, large-field of view (FOV) two-photon calcium imaging of ocular dominance in layer 4 of the visual cortex. Left: Schematic indicating volume dimensions and acquisition rate. Middle: Example FOV (GCaMP6s expression) of a single plane; "A", "P", "M", and "L" indicate anterior, posterior, lateral, and medial. Right: HLS map for ocular dominance. Hue: Eye-preference (contralateral: Blue; ipsilateral: Red); Lightness: ΔF/F; Saturation: Eye selectivity. Scale bars: 100 μm. For maps of all nine mice, see Supplementary Fig. 12c. **b** Left: ΔF/F response to drifting grating stimuli for three example cells (see **a**). "C": Contralateral, "I": Ipsilateral. Scale bar, vertical: 100% ΔF/F, horizontal: 10 s. Right: Inferred spiking activity of all visually responsive cells (n = 1834) in (**a**), sorted vertically by ODI and aligned horizontally to the preferred (P) direction (N: Null direction; blue/red: Contra/ipsi). Scale bar, vertical: 100 neurons, horizontal: 10 s. **c** Left: Local density of ipsilateral (top) and contralateral (bottom) eye preferring visually responsive neurons separately (across four-plane volume). Black line: lateral boundary of V1 (V1 is to the right). Black circles: Ipsi-cluster centers (see "Methods"). Right: All visually responsive neurons, color coded for ODI. White iso-ODI lines indicate ODI = 0 (solid) and ODI = 0.2 (dashed). Scale bar: 100 μm. **d** Histogram of ODI (for volume in

**c**). **e** Mean ODI as a function of distance to the three ipsi-cluster centers in (**c**) (individual ipsi-clusters in gray). Red line: sigmoid fit; the point of maximum inclination (red arrow) approximates the ipsi-cluster radius. **f** Preferred azimuth (top) and elevation (bottom) of the neurons shown in (**c**). Scale bar: 100 μm.
**a**–**f** Data of mouse M02. **g** Left: ODI as function of distance to ipsi-cluster centers. Black: Mean ± s.e.m., gray: Individual mice (n = 9). Right: Mean ODI inside ("In", 0–100 μm) and outside ("Out", 100–200 μm) ipsi-clusters (two-sided WMPSR test, W = 0, p = 0.004, n = 9 mice). **h** Same as (**g**) black line (mean ± s.e.m.) shows actual data ("D"), blue, green, and red lines show global randomization controls. Right: Mean (±s.e.m.) ODI "In" ipsi-clusters for real and shuffled data (two-sided Kruskal-Wallis test, $H_3$ = 17.7, p = 5.0 × 10⁻⁴, post hoc two-sided WMPSR test, **p < 0.01, n = 9 mice). Blue, "S": ODI values shuffled across neurons. Green, "U": XY coordinates of neurons resampled from uniform distribution. Red, "R": Ipsi-cluster centers randomly placed in FOV. **i** As (**h**), colored lines show local randomization controls in which the positions of pairs of neurons, spaced at a distance of 50 μm (dark red) to 250 μm (yellow), were swapped (two-sided Kruskal-Wallis test, $H_5$ = 16.8, p = 0.0048, post hoc two-sided WMPSR test, ns not significant, **p < 0.01, n = 9 mice).

relatively similar to its surround. However, the mean ODI near ipsi-clusters was just below zero (−0.07 ± 0.15 s.d.), indicating that many nearby neurons indeed responded preferentially to the ipsilateral eye. Furthermore, ODI increased with distance from the cluster center, showing that neurons outside the ipsi-clusters generally preferred the contralateral eye (Fig. 1e, g; see also Supplementary Fig. 4d).

Comparison with single neuron-resolution retinotopic maps revealed that the ipsi-clusters were located within binocular V1 (Fig. 1f; Supplementary Figs. 1, 2a–d). Individual ipsi-clusters varied in shape from being round to elongated, and occasionally had irregular features. All ipsi-clusters (average size: 162 μm, ± 26 μm s.d.; see "Methods") were substantially smaller than the extent of binocular V1, and in

most mice we detected multiple ipsi-clusters (2.7 ± 0.7 s.d. ipsi-clusters per mouse) within the ~50% of binocular V1 that our field of view covered on average.

We performed several global randomization procedures to test whether ipsi-clusters could occur by chance: neither when shuffling the ODI values across neurons, when randomly repositioning the ipsi-cluster centers, nor when repositioning neurons at random XY coordinates in the imaged region did we observe ipsi-clusters having ODI values comparable to the real data (Fig. 1h). Across individual trials in imaging sessions, we found virtually no effect of trial-to-trial fluctuations in single neuron ODI values on the detected positions of ipsi-clusters (Supplementary Fig. 3). This shows that ipsi-clusters, as observed, do not emerge from random spatial distributions of ipsilateral eye preferring cells.

Because our imaging field of view (FOV) was wider (~1.2 mm) than the medial-lateral extent of the binocular visual cortex (~0.8 mm), it could be that the effect in Fig. 1h, at least partly, reflected the monocularity of neurons outside the binocular visual cortex. To address this issue, we tested whether the ipsi-clusters were part of a fine-grained spatial organization for ocular dominance, smaller than the extent of binocular V1, or whether they could be explained by global effects like boundaries between binocular and monocular cortex. This was done by a local randomization control. By swapping the ODI values of pairs of neurons that were separated by a small distance (e.g., 50 μm), we randomized the local spatial distribution of ODI values while maintaining the larger-scale functional organization of monocular and binocular visual cortex (Supplementary Fig. 2e–h). This analysis showed that the ipsi-clusters indeed originated in the fine-grained arrangement of ipsilateral eye preferring neurons (Fig. 1i).

We quantified the fraction of ipsilateral and contralateral eye preferring neurons relative to ipsi-cluster centers and observed a similar pattern: The fraction of ipsilateral eye preferring neurons (ODI < 0) was larger near ipsi-clusters (Supplementary Fig. 4a–c; this also held for ipsilateral eye preferring neurons with ODI < −0.3, or for neurons responding significantly stronger to the ipsilateral compared to the contralateral eye; Supplementary Fig. 5). The increased fraction of ipsi-lateral eye preferring neurons likely resulted from increased neuronal responses (across all neurons, both ipsilateral and contralateral eye-preferring) to ipsilateral eye visual stimulation near ipsi-clusters (Supplementary Fig. 4e, left). Interestingly, contralateral eye stimulation did not result in reduced responses near ipsi-clusters (Supplementary Fig. 4e, right), suggesting overall stronger visual drive near ipsi-clusters. Accordingly, the overall binocular response amplitude in the separate binocular retinotopic mapping experiment was largest near ipsi-clusters (Supplementary Fig. 4f). Finally, investigating the ODI of ipsilateral eye and contralateral eye preferring neurons separately, we observed only a relatively minor shift towards more negative ODI values near ipsi-clusters (Supplementary Fig. 4g). These data together suggest that ipsi-clusters are characterized by locally increased ipsilateral drive, leading to an increase in the fraction of ipsilateral eye preferring neurons.

We excluded that the functional clustering of ocular dominance described here is a specific feature of how we preprocessed the imaging data, i.e., using Suite2P[26–28]. Suite2P calculates ROI fluorescence timeseries from the mean signal intensity within the ROI, and subtracts the mean signal intensity in the local neuropil multiplied by a factor of 0.7 (see "Methods"). The CaImAn package[29] is based on a different approach, using non-negative matrix factorization to extract cellular ROIs and their fluorescence timeseries from the mixed foreground and background signals that make up fluorescence imaging data (see "Methods"). However, because CaImAn is optimized to extract sparse neuronal activations from dense background signals, neuronal responses can appear more selective for stimuli (e.g., result in more extreme ODI values in our data) than with Suite2P. Nevertheless, the presence and location of ipsi-clusters, and the comparison with shuffle

controls were similar to what we observed using Suite2P (Supplementary Fig. 6).

Finally, we confirmed our findings in a different mouse line using virus-mediated (AAV2/1) expression of a red-shifted calcium indicator (jRGECO1a) in the right hemisphere of Scnn1a-Tg3-Cre transgenic mice[30], thus limiting expression in V1 to cortical layer 4 (see "Methods"). Notably, in this mouse line neuropil signals would only originate from jRGECO1a expressing layer 4 neurons and not from, e.g., layer 5 dendritic bundles in the layer 4 neuropil[16]. Despite small quantitative differences, which likely resulted from a less homogeneous distribution and overall smaller number of neurons expressing the calcium indicator, we confirmed the overall finding of ipsi-clusters in binocular V1 (Supplementary Fig. 7).

## L4 ipsi-clusters extend vertically into cortical layers 2/3 and 5

Having found clusters of ipsilateral eye preferring neurons in layer 4 of mouse binocular V1, we asked whether these ipsi-clusters extended vertically into other cortical layers, i.e., show a column-like arrangement. In each animal (n = 9), we acquired an imaging volume spanning 360 μm of cortex from upper L2/3 (170 μm below the pial surface) to upper L5 (530 μm; 37 planes, spaced 10 μm apart; see Fig. 2a). The volume was constructed from 12 individual four-plane imaging stacks, which were acquired in random order (see "Methods"). Each L2/3-L5 volume contained on average 24366 (±6384 s.d.) visually responsive neurons, with 18305 (±5879 s.d.) neurons preferring the contralateral eye and 6061 (±2600 s.d.) preferring the ipsilateral eye and a mean ODI of 0.27 (±0.11 s.d.) across all imaged cortical layers.

In most mice, pixelwise ODI maps across different cortical depths showed a clear similarity in the overall patterns of ipsilateral and contralateral eye dominated regions (Fig. 2b, left; Supplementary Fig. 8). Using the method described above, we identified centers of ipsi-clusters in the L4 subvolume spanning 350 μm to 430 μm below the pial surface (Fig. 2b, right; see "Methods"). The ODI of neurons in a small column (<100 μm range around ipsi-cluster centers, <200 μm diameter) above and below the L4 ipsi-cluster centers was significantly lower (more ipsilateral) than the ODI of neurons further outwards (range: 100–200 μm, diameter: 200–400 μm; Fig. 2c; Supplementary Figs. 9 and 10).

As within layer 4, the global and local shuffle controls showed that the low ODI in the column above and below the L4 ipsi-cluster centers did not occur by chance (Fig. 2d, e), nor did it reflect the division of visual cortex in monocular and binocular regions, as the regions of vertical clustering were substantially narrower than the extent of binocular V1 (see Supplementary Figs. 8–10). Similar to the within-L4 patterns of ipsi-clusters, the strength of the vertical alignment varied across mice (see Supplementary Figs. 8 and 10 gray lines, for single mouse data across cortical layers). Thus, the ipsi-clusters we detected in cortical layer 4 extended vertically, in a column-like fashion, at least into cortical layers 2/3 and 5.

In order to directly quantify to which degree the spatial organization of ODI in layer 4 aligned across cortical layers, we constructed high-resolution ODI-maps based on cellular ODI values for four depth ranges (corresponding to upper and lower L2/3, L4 and upper L5; see "Methods"; Supplementary Fig. 11a). The similarity in ODI patterns between L4 and other layers was calculated as the two-dimensional cross-correlation between ODI-maps (Fig. 2f; Supplementary Fig. 11a, b). The peaks of nearly all cross-correlation maps were close to the origin (0,0), confirming the alignment of the overall ODI pattern across layers (Fig. 2g, black circles; Supplementary Fig. 11c, d). In comparison, the locations of cross-correlation peaks for ODI maps calculated using globally shuffled ODI values (shuffled ODI), and using imaging planes from altogether different animals (shuffled planes) were randomly distributed (Supplementary Fig. 11c, d).

Comparing cross-sections of cross-correlation maps with those of the local shuffle control (swapped ODIs at 200 μm) shows a narrow

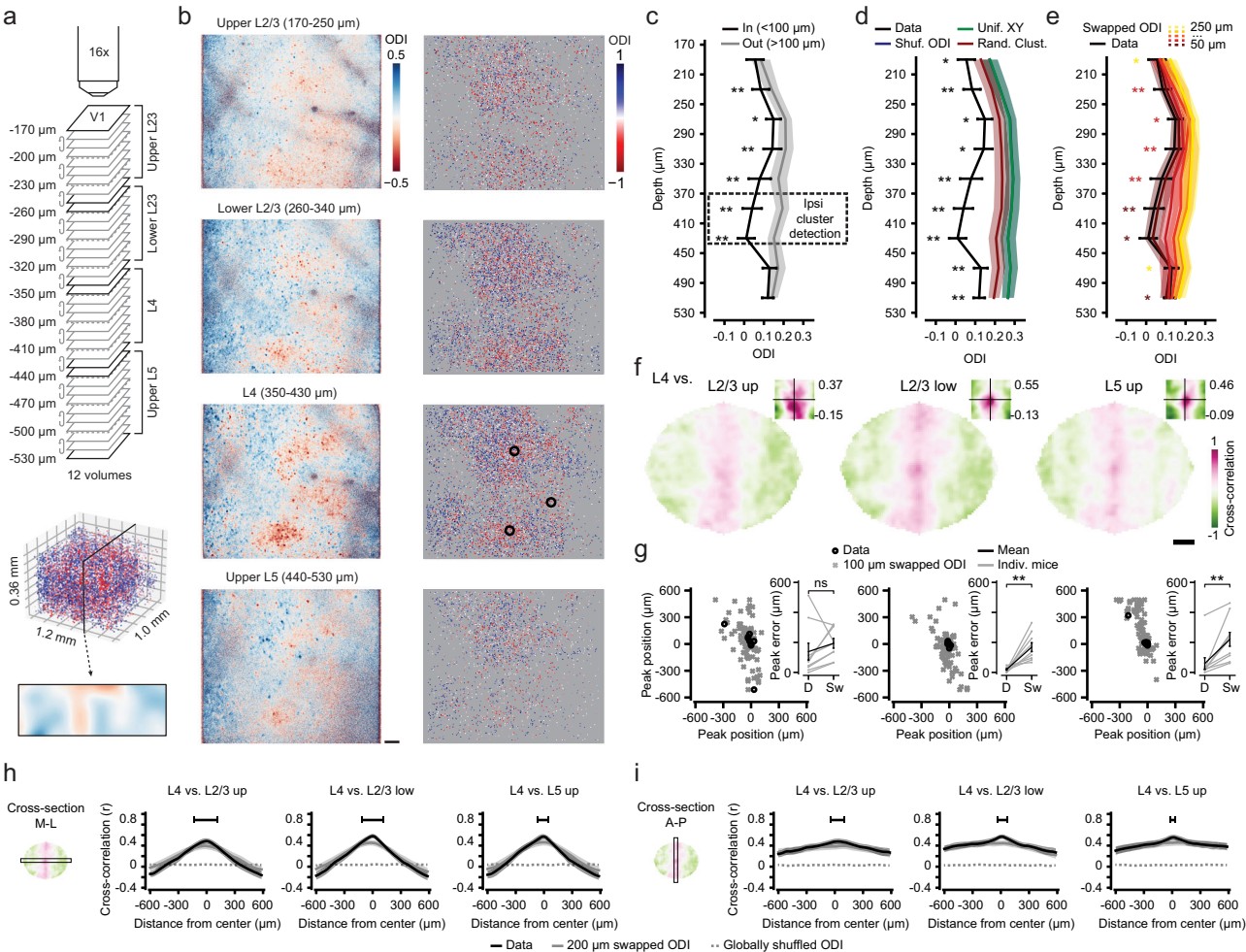

**Fig. 2 | A column-like organization for ocular dominance in mouse visual cortex layers 2/3, 4 and 5. a** Approach for recording ocular dominance across a cortical volume. Top: Schematic showing 12 multilevel imaging stacks (acquired in random order) resulting in 37 uniquely imaged planes, spaced 10 μm apart in depth. Middle: ODI of all visually responsive neurons in an example volume (*n* = 22898; color bar: **b** right). Bottom: Projection of ODI across a 100 μm thick vertical slice (color bar: **b**, left). **b** Left: Pixelwise ODI maps using imaging data combined across multiple imaging planes spanning four depth ranges (see **a**). Scale bar: 100 μm. Right: ODI of all visually responsive neurons across the same depth ranges. Black circles: Ipsi-clusters. **a**, **b** Data of mouse M02. **c** Mean (±s.e.m. across 9 mice) ODI "In" (<100 μm; black) and "Out" (100–200 μm; gray) of ipsi-clusters, detected in layer 4 (dashed box), across nine depth bins (tick marks indicate bin-edges). **d**, **e** As (**c**) real data (mean ± s.e.m., black) "In" ipsi-clusters versus global and local randomization controls. **d** Global randomization control. Blue: Shuffled ODI. Green: Uniformly resampled XY coordinates. Red: Randomly placed ipsi-cluster centers. Note: Green line overlays blue line. **e** Local randomization control. Dark red to yellow mark swap distances from 50 μm to 250 μm. **c**–**e**, Statistical comparison of ODI "In" ipsi-cluster centers versus all controls ("Out", local and global shuffles), Kruskal-Wallis tests per depth bin, *p* < 0.05, corrected for 9 comparisons; post hoc two-sided WMPSR tests, *\*p* < 0.05, \*\**p* < 0.01, in **e** color coded for smallest significant swap distance, *n* = 9

mice. **f** 2D cross-correlation of L4 ODI maps with those of L2/3 and L5 (see "Methods"; see Supplementary Fig. 11a; Data of mouse M02). Insets show cropped map centers with the cross-correlations scaled to individual minima and maxima (see values on the right, scale identical to full cross-correlation map). Scale bar: 200 μm. **g** Spatial position of the cross-correlation peak (real data, "D", black) compared to locally randomized data ("Sw", swapped ODIs at 100 μm, 10 repeats, gray). Left: Upper L2/3, middle: Lower L2/3, right: Upper L5. Inset shows the peak error (mean ± s.e.m.), i.e., the Euclidian distance between the detected peak and the center of the cross-correlation map (L2/3 up: Two-sided WMPSR test, *W* = 10, *p* = 0.16; L2/3 low: Two-sided WMPSR test, *W* = 0, *p* = 0.004; L5 up: Two-sided WMPSR test, *W* = 0, *p* = 0.004; *n* = 9 mice; ns not significant, \*\**p* < 0.01). **h** Left: Schematic showing a cross-section (box) of a cross-correlation map, orthogonal to the anterior-posterior (long) axis of binocular V1. Right three plots: Cross-correlation (mean ± s.e.m. across mice) along the cross-section. Black: Real data. Gray solid line: Local shuffle control (swapped ODIs at 200 μm). Gray dotted line: Global shuffle control (shuffled ODI). Statistical testing: Linear mixed-effects model (see "Methods"), interaction effect of condition (data vs. local shuffle) and measurement (distance from center), black bar above data indicates the range for which *p* < 0.01. **i** As (**h**), but for cross-sections of the cross-correlation map parallel to the anterior-posterior (long) axis of binocular V1.

range of alignment across depth ranges in which the cross-correlation exceeds that of the local shuffle control (Fig. 2h, i; Supplementary Fig. 11c, d). Note that the cross-correlation resulting from vertical alignment of the much larger binocular region of the primary visual cortex (V1b) can be seen in both real data and the local shuffle control as a broader "shoulder" of positive correlation values. In accordance, the cross-correlation peaks for ODI maps calculated using locally shuffled ODI values (ODIs swapped at 100 μm) were positioned close

to the center on the x-axis, likely reflecting the overall binocular-monocular gradient along this dimension (Supplementary Fig. 11c, e). Crucially, along the vertical image axis (y) there was much less of a coarse gradient in ODI values, and local randomization prevented the cross-correlation peaks to position near the center (Supplementary Fig. 11c, e). This indicates that the alignment of ODI patterns across cortical layers depends on the fine-grained functional organization of cellular ocular dominance.

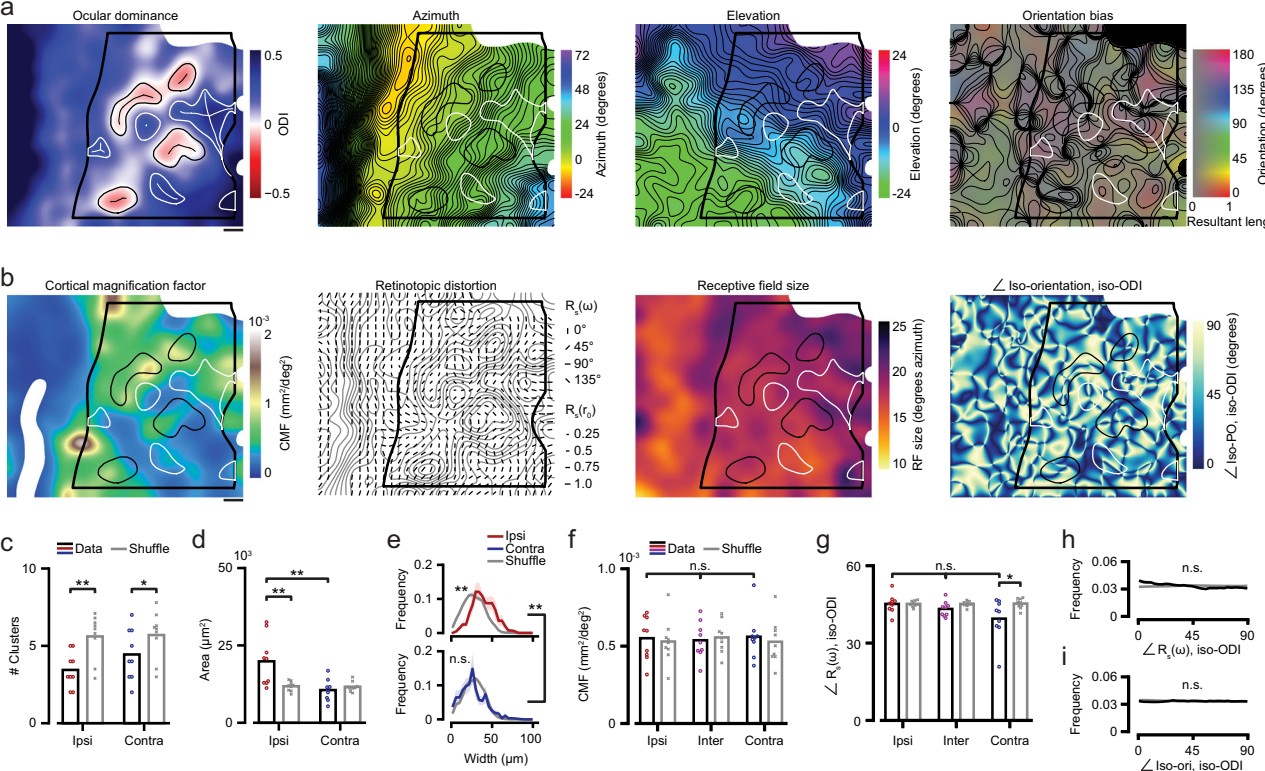

**Fig. 3 | Ipsi-clusters in layer 4 are non-randomly arranged, but show no spatial relationships with other feature maps. a** Smoothed feature maps of L4 single neuron receptive field properties (example mouse M02). Ocular dominance: Thick black line shows V1b boundary (see "Methods"). Boundaries and skeletons of ipsi-clusters and contralateral eye dominated regions are marked with black and white lines, respectively (ipsi-clusters are marked in black). Scale bar: 100 μm. Preferred azimuth, preferred elevation: Thin black lines indicate iso-azimuth/iso-elevation at 1° spacing. Orientation bias: The angle (hue) and length of the local resultant vector. Thin black lines indicate iso-orientation at 22.5° spacing. **b**, As **a**, for the cortical magnification factor (CMF); the retinotopic distortion vector $R_s$[37] (short black lines, see legend), overlaid on iso-ODI lines in gray (0.05 spacing); for receptive field size; and for the angle between iso-ODI and iso-orientation gradients. Scale bar: 100 μm. **c** Number of detected ipsi-clusters and contralateral eye dominated regions (using ODI-map method, see Supplementary Fig. 12a, b) for real (red, blue, black) and ODI-shuffled data (gray; Two-sided WMPSR test; ipsi vs. shuffle: $W = 1$, $p = 0.008$; contra

vs. shuffle: $W = 1$, $p = 0.017$; $n = 9$ mice. **d** As (**c**), but for the size (area in μm²) of the clusters (Two-sided WMPSR test; ipsi vs. contra: $W = 1$, $p = 0.008$; ipsi vs. shuffle: $W = 1$, $p = 0.008$; $n = 9$ mice). **e** Distribution of the widths (mean ± s.e.m., see "Methods") of ipsi-clusters and contralateral eye dominated regions (red, blue) versus shuffled ODI data (gray; Two-sided WMPSR test; ipsi vs. contra: $W = 1$, $p = 0.008$; ipsi vs. shuffle: $W = 0$, $p = 0.004$; $n = 9$ mice). **f** Cortical magnification factor for ipsi-clusters (red), contra- (blue), and intermediate regions (pink) regions (ipsi, inter, contra: Kruskal-Wallis test, $p > 0.05$), versus shuffled ODI data (gray). **g** As (**f**), for the angle between the retinotopic distortion vector ($R_s(\omega)$) and iso-ODI angle (ipsi, inter, contra: Kruskal-Wallis test, $p > 0.05$; Two-sided WMPSR test; contra vs. shuffle: $W = 4$, $p = 0.027$; $n = 9$ mice). **c**, **d**, **f**, **g**, Bars show mean across mice. **h** Histogram of the angle between the retinotopic distortion vector ($R_s(\omega)$) and iso-ODI angle across V1b (shuffle ODI data in gray). **i** As (**h**), but for the angle between iso-ODI angle and iso-orientation angle. **h**, **i** Data vs. shuffle: Two-sided WMPSR test, $p > 0.05$.

## L4 ipsi-clusters show a non-random spatial arrangement, but no relationship with the retinotopic map

A well established feature of the visual cortex of higher mammals is that the pattern of ocular dominance columns is spatially related to other feature maps (e.g., for orientation, spatial frequency and retinotopy[1,7,31–33]). These specific arrangements on the cortical surface likely serve to ensure uniform coverage of feature representations across the visual field[1]. While mice reportedly do not have feature maps for preferred orientation and spatial frequency, they do have a retinotopic map[11,34], which, like in higher mammals, is not uniform and shows anisotropies[35]. Having here characterized a column-like organization for ocular dominance in mice, we can now ask whether feature maps in mice show similar spatial relationships.

In order to assess the relationship between the maps for retinotopy and ocular dominance (Fig. 3a), we implemented an algorithm that identifies the outlines of ipsi-clusters (Supplementary Fig. 12a), as well as contralateral eye dominated cortical regions (Supplementary Fig. 12b). This approach allowed us to quantify a number of geometric and functional properties of ipsi-clusters (see Table 1 and Supplementary data 1). Generally, the visual cortex contained fewer ipsi-clusters than expected

from randomized maps based on shuffled ODI values (Fig. 3c), ipsi-clusters were more often located in the representation of the frontal-upper visual field (Supplementary Fig. 12d, e) and ipsi-clusters were larger in size than expected by chance (Area, Fig. 3d; Width, Fig. 3e). Furthermore, neurons in ipsi-clusters were slightly more orientation selective and slightly less direction selective (Supplementary Fig. 12f, g).

First, we tested the hypothesis that accommodating two visual channels (i.e., inputs from the ipsilateral and the contralateral eye), requires more cortical space. We computed, from the retinotopic map, the cortical magnification factor (CMF)[36], which describes how much cortical area is dedicated to processing a square degree of visual field (Fig. 3b). Because ipsi-clusters are much more binocular than contralateral eye preferring regions, we expected a larger CMF in ipsi-clusters[1,31]. However, we observed no differences in CMF between ipsi-, contra- and intermediate regions or with shuffle controls (Fig. 3f), and related to this, we did not find significant differences in receptive field size (Supplementary Fig. 12h). Nor did we observe any significant correlation between the overall CMF or overall receptive field size in each mouse, and the number of ipsi-clusters, or ODI in ipsi-clusters in those animals (Supplementary Fig. 13).

**Table 1 | Properties describing ipsi-clusters and their interaction with the retinotopic map**

| Mouse | # ipsi clusters | Area (mm²) | Length (µm) | Width (µm) | # ipsi neurons | # contra neurons | ODI (ipsi) | ODI (V1b) | CMF (ipsi) (mm²/deg²) | CMF (V1b) (mm²/deg²) | Rs(ω) angle |
|---|---|---|---|---|---|---|---|---|---|---|---|
| M01 | 2 | 0.031 | 230 | 44 | 118 | 85 | −0.030 | 0.167 | 0.00032 | 0.00047 | 43.3 |
| M02 | 4 | 0.023 | 128 | 48 | 98 | 58 | −0.147 | 0.127 | 0.00060 | 0.00048 | 39.8 |
| M03 | 4 | 0.021 | 150 | 35 | 49 | 48 | 0.000 | 0.319 | 0.00045 | 0.00059 | 52.2 |
| M04 | 2 | 0.033 | 324 | 34 | 122 | 90 | −0.091 | 0.239 | 0.00060 | 0.00054 | 42.9 |
| M05 | 5 | 0.013 | 54 | 44 | 54 | 35 | −0.104 | 0.086 | 0.00047 | 0.00058 | 45.4 |
| M06 | 3 | 0.014 | 37 | 47 | 32 | 31 | −0.071 | 0.246 | 0.00043 | 0.00037 | 47 |
| M07 | 5 | 0.013 | 133 | 30 | 22 | 46 | 0.145 | 0.280 | 0.00072 | 0.00065 | 46.3 |
| M08 | 3 | 0.021 | 202 | 34 | 105 | 81 | −0.036 | 0.146 | 0.00070 | 0.00073 | 46 |
| M09 | 3 | 0.011 | 89 | 35 | 38 | 16 | −0.191 | 0.008 | 0.00069 | 0.00051 | 44.8 |

Each row shows the mean data for a single mouse. # ipsi clusters: The number of clusters detected using the geometry method (Supplementary Fig. 12a, b). Area: Cortical area covered by ipsi-clusters. Length: The length of the ipsi-cluster skeleton axes (see Supplementary Fig. 12a, b). Width: Average minimum distance from ipsi-cluster skeleton axes to cluster boundary. ODI (ipsi): ODI of cells in ipsi-clusters. ODI (V1b): ODI of cells across binocular V1. CMF (ipsi): Cortical magnification factor in ipsi-clusters. CMF (V1b): Cortical magnification factor across binocular V1. Rs(ω) angle: The angle of the retinotopic distortion vector relative to the iso-ODI orientation in ipsi-clusters.

In higher mammals, cortical retinotopy is intricately linked to the pattern of ocular dominance columns. Specifically, the direction of maximum ocular dominance segregation (running orthogonal to the columns) aligns with the axis of the slowest retinotopic gradient, which is not only observed globally across the entirety of V1, but also as local deformations in the retinotopic map on a spatial scale roughly matching the size of the columns[37,38]. Because we observed inhomogeneities in the spacing of iso-azimuth and elevation lines in mouse V1b, we asked whether these deformations were related to the pattern of ipsi-clusters in a similar way. However, the vector ($R_s$) describing the magnitude ($R_s(r_0)$) and direction ($R_s(ω)$) of local retinotopic distortion[37] was not different between ipsi-, contra-, and intermediate regions (Fig. 3g). Contralateral eye preferring subregions did show a slightly smaller angle between iso-ODI and $R_s(ω)$, compared to shuffle control. This could be related to contralateral eye preferring regions being located closer to the binocular-monocular boundary of V1 (see Supplementary Fig. 12d). This boundary runs 'vertically' in our maps, roughly aligning with the iso-elevation gradient, which is overall shallower than the azimuth gradient and therefore biases the distortion vector in the same direction[34]. However, the angle between iso-ODI and $R_s(ω)$ across V1b was not different from the shuffle control (Fig. 3h), indicating the absence of an overall spatial relation between the ocular dominance and retinotopic maps.

Finally, while in mice there is no orientation preference map as observed in higher mammals, there have been reports of local biases in preferred orientation[16,17,19]. Therefore, we additionally investigated the angle between iso-ODI and iso-orientation gradients[1,32,33]. Note that any local biases in orientation preference in mouse V1 were very weak, having a resultant length of ~0.12 (shuffled maps had mean resultant lengths of ~0.10), and thus should not be interpreted as evidence for an orientation preference map in mice. Indeed, in accordance with the virtual absence of such a map, the distribution of angles between iso-orientation gradients and iso-ODI gradients was completely flat, and not different from that of shuffled data (Fig. 3i).

In summary, we find that mouse visual cortex contains a functional organization for ocular dominance that has a column-like vertical extent, but that, in mice, these structures lack interactions with other feature maps, at least, to the level of detail that our data can provide.

## Discussion

Finding column-like structures for ocular dominance in the minute mouse binocular visual cortex provides a new opportunity for investigating the general question of which factors determine whether a cortex has columns or not. Experimental[13,39,40] and theoretical[41] studies have argued that columnar architectures in the visual cortex are only found in certain mammalian orders, such as primates, carnivorans, ungulates, scandentians and diprotodonts, but not in others. Rodent visual cortex, in particular, has been thought to lack columnar organizations. Our and others' recent findings of ocular dominance (rat[21–23]) and ON/OFF (mice[18]) domains show that this distinction based on taxonomy does not hold. For another prominent columnar system in the visual cortex, the orientation preference map, the situation is less clear. So far, no such map has been found in any rodent, and we similarly do not find evidence for an orientation preference map within binocular V1. However, other preliminary data indicate the possibility of a large-scale organization for preferred orientation across all of mouse visual cortex[19], and orientation-minicolumns have been found in several rodent species[16,17,40,42]. Importantly, orientation and direction preference maps have been observed in another part of the mouse visual pathway, the superior colliculus[43–46] (see however ref. [47]). Thus, circuits in the visual system of the mouse and rat are in principle capable of organizing into column-like structures. Studies in a larger variety of rodent species might reveal whether their visual cortex can also hold a "proper" orientation map, like those found in cats and primates[7,9].

The existence of column-like structures for ocular dominance in mice raises the question of whether this functional organization is directly comparable to what has been observed in higher mammals. While we similarly observe a vertical arrangement of neurons having the same eye-preference, notably, the pattern of ipsilateral eye dominated regions in mice does not resemble stripes or bands, and the ipsi-clusters appear far smaller[38] and less homogenous[48] than in other species (see also Table 1). Moreover, because the average ODI in ipsi-clusters is only slightly below zero, the ipsilateral eye preferring clusters could be argued to reflect regions for binocular processing within a contralateral eye dominated visual cortex. One explanation could be that, in rodents, ipsilateral eye preferring columns are more closely related to callosal patches[23,49,50], although these structures cannot be considered equivalent[51,52]. Our observation of slightly increased -binocular- visual drive in ipsi-clusters (Supplementary Fig. 4f) could reflect the additional visual drive of callosal axons targeting these patches in mice. Alternatively, the quantitative differences of features describing vertically extending structures across different species could merely reflect differences in experimental approaches. The prominent contralateral eye bias we and others observe in the mouse could partially be an effect of anesthesia[15]. Additionally, our method (large field of view calcium imaging) likely provides a different sample of visual cortex neurons, and of their activity, in comparison to methods like electrophysiology or c-Fos mapping. Thus, at this point it would be too early to conclude whether the column-like organization for ocular dominance in mice is fundamentally the same structure as observed in other species.

Apart from taxonomy, factors like overall visual cortex size[53], the absolute number of neurons[54], the retino-thalamo-cortical mapping

ratio[55,56] and the visual sampling density by geniculocortical afferents in visual cortex[38,57] have all been put forward in theoretical work to explain the absence of a functional architecture and columnar structures in a visual cortex as small as that of the mouse. While some of these studies refer to a columnar architecture in general, or orientation columns specifically, others[54,58] explicitly predict that mouse visual cortex does not have a spatial organization for ocular dominance, contrary to what our experiments show. Our finding will help improving future models of visual cortex functional architecture.

In cats and monkeys, during early development, ocular dominance columns have been shown to gradually emerge from initially intermingled thalamic axons driven by one or the other eye[59,60]. This process is generally thought to be governed by neuronal activity, either visually driven or internally generated[61], and serves as a prime example for activity dependent development in the nervous system. In rat V1, ocular dominance domains align with ipsilateral eye thalamocortical projections and emerge before eye opening[23,52], suggesting that also in rodents axon sorting might contribute to the formation of column-like structures. A recent model provides further insight into the anatomical prerequisites for this process, showing that the segregation of ocular dominance in V1 emerges when afferent sorting can be operated on iso-retinotopic regions large enough to allow a significant number of afferents to sample the same point in visual space[57]. The required oversampling of visual space predicts that ocular dominance columns interact with the retinotopic map[37,38]. However, we were unable to confirm such interactions in the mouse, thus, other mechanisms might also contribute to the formation of the cortical structures we observed. One alternative (or additional) explanation points to molecular axon guidance cues being an important factor for the formation of ocular dominance columns[62,63]. Research on this topic has not progressed much recently, since the experimental work was largely performed in ferrets and cats. These species do not lend themselves easily to genetic interventions, which may be crucial to elucidate the molecular nature of such putative guidance cues.

Our finding expands the range of mammalian species, across very diverse orders, which display a column(-like) organization for ocular dominance. That such structures are apparently rather the rule than an exception is in stark contrast to the lack of a clear hypothesis of what their function for visual processing is, if there is one at all[64]. Possible explanations range from intracortical wirelength minimization[41,54,65], over merely being an epi-phenomenon created by the activity dependent wiring of cortical circuits[66], to a not very clearly spelled out function for binocular integration and stereoscopic depth perception[67]. A general way to probe the function of a structure in the brain is to remove it, and test for ensuing changes in neuronal processing and behavior. This appears very difficult for columnar organizations, since eliminating the architecture without massively affecting cortical circuitry altogether seems impossible. There is, however, an experiment by nature, which might prove helpful for answering this question. Squirrel monkeys show a "capricious" expression of ocular dominance columns in their visual cortex, ranging from fully developed columns in some animals to nearly complete absence in others[68].

While we have not systematically explored the variability in the degree of columnar organization in our mouse data, there are clear and reproducible differences between individual animals (see e.g., maps in Supplementary Fig. 12c). While this variability could originate from subtle differences in experimental conditions, it might as well result from natural variability in processes that give rise to column-like structures for ocular dominance and thereby provide a new opening for studying the role of columnar organization in supporting cortical computation. For instance, recent experiments have shown that mouse visual cortex contains many neurons sensitive to binocular disparities[42,69], and that mice make use of such binocular cues for judging distances[70–72]. Relating the degree of columnar organization for ocular dominance to neuronal or behavioral signs of binocular depth perception might reveal whether the arrangement of cortical neurons into eye specific column-like patterns is relevant for these important visual system functions.

## Methods

### Animals

All experiments were conducted following the institutional guidelines of the Max Planck Society and the regulations of the local government ethical committee (Beratende Ethikkommission nach §15 Tierschutzgesetz, Regierung von Oberbayern). Nine adult mice (6 female, 3 male; 4–5 months of age during data acquisition) genetically expressing the calcium indicator GCaMP6s in excitatory neurons (B6;DBA-Tg(tetO-GCaMP6s)2Niell/J[24], JAX stock #024742; back-crossed for at least seven generations to C57Bl/6NRj) crossed with B6.Cg-Tg(Camk2a-tTA)1Mmay/DboJ[73] (JAX stock #007004; maintained on a mixed background of C57BL/6NRj and C57BL/6J) and seven adult Scnn1a-Tg3-Cre mice (4 female, 3 male; 4 months of age during data acquisition; B6;C3-Tg(Scnn1a-cre)3Aibs/J[30], JAX stock #009613; kept on a mixed background of C57BL/6 and C3H) were housed in small groups, or individually in case of inter-male aggression, in large cages (GR900, Tecniplast) containing a running wheel, a tunnel and a shelter. The animals were kept on a reversed day/night cycle with lights on at 22:00 h and lights off at 10:00 h. Ambient temperature (21.0 ± 0.7 °C) and humidity (63 ± 2%) were kept constant. Food and water were available *ad libitum*.

### Surgery

Animals were anesthetized with a mixture of fentanyl (0.05 mg/kg), midazolam (5.0 mg/kg) and medetomidine (0.5 mg/kg) in saline (injected i.p.; FMM in short). For analgesia, carprofen (5.0 mg/kg) was injected s.c. and lidocaine (0.2 mg/ml) was applied topically. A head bar and a 4 mm diameter cranial window (cover glass, #1 thickness) were implanted[74,75]. The cranial windows were placed over the visual cortex of the left hemisphere in GCaMP6s transgenic mice and the right hemisphere in Scnn1a-Tg3-Cre mice. In Scnn1a-Tg3-Cre mice, AAV2/1.Syn.Flex.NES-jRGECO1a.WPRE.SV40 (titer: 2.6 × 10¹³; a gift from Douglas Kim & GENIE Project, Addgene viral prep #100854-AAV1) was pressure-injected using a glass micropipette at ~400 μm depth (200–250 nl per injection), at 4–6 locations spanning binocular V1 (identified using intrinsic optical signal imaging)[76]. Following surgery, animals received antagonists (1.2 mg/kg naloxone, 0.5 mg/kg flumazenil, and 2.5 mg/kg atipamezole in saline, injected s.c.). Post-operative analgesia (5.0 mg/kg carprofen injected s.c.) was given for two subsequent days. In a subset of mice, small patches of bone-growth under the window were removed in a second surgery.

### Imaging

In vivo calcium imaging was performed using a customized, commercially available Bergamo II (Thorlabs, Germany) two-photon laser scanning microscope[77] with a pulsed femtosecond Ti:Sapphire laser (Mai Tai HP Deep See, Spectra physics), running Scanimage 4[78]. GCaMP6s[79] was excited using a wavelength of 940 nm, and jRGECO1a[80] using 1050 nm. Lowpass (720/25 nm; Semrock, USA) and bandpass (GCaMP6s: 525/50-25 nm; jRGECO1a: 607/70-25 nm; Semrock, USA) filtered emitted fluorescence was detected with two GaAsP detectors (Hamamatsu, Japan). Laser power under the objective ranged from 15 to 45 mW, depending on the depth of imaging (upper L2/3 to L5). The field of view size of a single-plane image was 1192 × 1019 μm (XY; 1024 × 1024 pixels). Four-plane volumes were acquired at 3.8 Hz per plane using a 16× objective (0.8 NA; Nikon) attached to a piezo electric stepper (Physik Instrumente, Germany).

During imaging, animals were lightly anesthetized with FMM (see above) and kept warm on a heat pad (closed loop temperature controller set to 37 °C). Eyes were kept moist using eye drops (Oculotect).

Visual stimuli were presented on a gamma corrected and curvature corrected[81] computer monitor. Ocular dominance and orientation tuning were assessed by presenting full screen, 100% contrast, square wave drifting gratings (spatial frequency: 0.04 cycles/degree; temporal frequency: 1.5 cycles/second) moving in one of eight possible directions and presented to the ipsilateral and contralateral eye separately using motorized eye-shutters. Stimulus presentation lasted 5 s, followed by a 6 s intertrial interval (ITI). Eye shutter switches were done 6 s post stimulus offset and were followed by a 10 s post-switch interval, after which the next ITI started. Trials were presented in blocks of 16 stimuli, containing two blocks of eight movement directions, that is, one block for each eye. The eight directions were randomized per trial block, as the order of eye blocks. Each full block of 16 stimuli was presented 10 times in L4 experiments and five times in L2/3-L5 experiments.

For mapping retinotopy, drifting gratings having one of four (cardinal) directions were presented in subsections of the visual field (patches) measuring 26 × 26 retinal degrees (azimuth × elevation). The patches were centered on −48, −24, 0, 24, and 48 degrees azimuth and −24, 0 and 24 degrees elevation. Individual trials were grouped into blocks containing all combinations of 15 patches (visual field partitions) and 4 movement directions (thus 60 unique stimuli) in randomized order. Four blocks were presented per experiment, with each stimulus presentation lasting 4 s and each ITI lasting 5 s.

## Response maps

To visualize cortical response properties throughout a single field of view, we calculated pixel-wise ODI (ocular dominance index) and HLS (Hue, Lightness, Saturation) maps (see Fig. 1a, right panel, Fig. 2b, left panels, and Supplementary Figs. 1a, 7a and 8a, c, e, g, i for examples). First, stimulus fluorescence (F) images were produced by averaging images across all trials of each stimulus (from stimulus onset to stimulus offset) and a baseline F image was created by similarly averaging images acquired during the intertrial interval periods (from 0.9 ITI length before stimulus onset until stimulus onset). The stimulus and baseline F images were smoothed using a median filter (disk kernel, 1 pixel radius). Next, ΔF/F response maps were created for each stimulus by subtracting the baseline F image from the stimulus F image and dividing the result by the baseline F image. Negative and infinite values (caused by division by zero) were set to zero. HLS maps for ocular dominance were created by assigning for each pixel: (1) The color, hue (H), reflecting which eye resulted on average in the largest ΔF/F response (red for ipsilateral and blue for contralateral), (2) the brightness, lightness (L), reflecting the amplitude of the largest ΔF/F response, and (3) the saturation (S), reflecting the selectivity for either eye (similar to the ODI). ODI maps were created similarly, but here only the ODI value (see Eq. 1 below) per pixel was encoded according to a colormap (see legend next to maps).

## Image processing and source extraction

Volumetric imaging stacks were analyzed per plane using customized scripts (see "Code availability") utilizing Suite2p version 0.7[26–28]. Image preprocessing consisted of dark-frame subtraction, line shift correction and image registration (rigid and non-rigid, with the following parameters, "two_step_registration": False, "nimg_init": 100, "maxregshift": 0.1, "subpixel": 10, "smooth_sigma_time": 10, "smooth_sigma": 1.15, "th_badframes": 1.0, "pad_fft": False,"do_nonrigid": True, "block_size": [128, 128], "snr_thresh": 1.2, "maxregshiftNR": 10). Source extraction and spike inference was done using the Suite2p algorithm (with these parameters, "tau": 1.5, "sparse_mode": False, "diameter": (Y,X) aspect-ratio-corrected to 12 μm, "spatial_scale": 0, "connected": True, "nbinned": 5000, "max_iterations": 20, "threshold_scaling": 1.0, "max_overlap": 0.75, "high_pass": 100, "inner_neuropil_radius": 2, "min_neuropil_pixels": 350, "allow_overlap": False, "chan2_thres": 0.65, "baseline": 'maximin', "win_baseline": 60.0, "sig_baseline": 10.0,

"prctile_baseline": 8.0, "neucoeff": 0.7) and all further analyses were performed on the resulting inferred spiking activity per neuron.

Potential non-cell specific signals in fluorescence traces (i.e., neuropil contamination) was corrected for by subtracting 70% of the local neuropil signal from each ROI fluorescence trace. We verified that the ROI footprints were similar in shape across cortical layers (Supplementary Fig. 14), excluding the possibility that data in upper cortical layers would, in part, consist of layer 5 dendritic signals. For L2/3-L5 data sets, the last and first plane of each pair of consecutive stacks were imaged at the same cortical depth, but data of only one of those planes (last) were used for analysis. Of neurons that occurred at the same XY position (centroid within a 6 μm diameter circle around another centroid) in consecutive planes, only the most significant tuned neuron (lowest p-value for stimulus tuning, see below) was kept.

CaImAn source extraction[29] was run using the following motion correction parameters, "strides": (48, 48), "overlaps": (24, 24), "max_shifts": (8,8), "max_deviation_rigid": 4, "pw_rigid": True, "use_cuda": True; the following parameters for source extraction and deconvolution, "decay_time": 1.49, "p": 2, "gnb": 7, "merge_thr": 0.839, "rf": 26, "stride_cnmf": 10, "K": 59, "gSig": [4, 4], "method_init": 'greedy_roi', "ssub": 1, "tsub": 1; and the following parameters for component evaluation, "min_SNR": 2.25, "rval_thr": 0.784, "cnn_thr": 0.9077, "cnn_lowest": 0.232. Inferred spike activity traces of extracted sources were further processed in the same way as described for the Suite2p data processing pipeline.

## Tuning curve analysis

For each neuron and each trial, an average inferred spike response was calculated by subtracting the average inferred spike activity during baseline (1.8 s before stimulus presentation until stimulus onset; seven imaging frames) from the average inferred spiking activity during stimulus presentation. The trial-wise average inferred spike responses were grouped by stimulus features (eight directions × two eyes, or five azimuths × three elevations). We determined whether a neuron was significantly tuned to one or more stimulus features using a Kruskal-Wallis test, testing for differences in average inferred spike responses across stimuli (the alpha was set to 0.05). Throughout the manuscript, such stimulus-tuned neurons are referred to as "visually responsive neurons". For each visually responsive neuron recorded in ocular dominance and orientation tuning sessions, we quantified the ocular dominance index (ODI; Eq. 1) using the average inferred spiking response (R) to the preferred direction (pref dir) of each eye (ipsi or contra).

$$\text{ODI} = \frac{R_{\text{pref dir, contra}} - R_{\text{pref dir, ipsi}}}{R_{\text{pref dir, contra}} + R_{\text{pref dir, ipsi}}} \tag{1}$$

In addition, we calculated the preferred orientation and direction from the stimulus that elicited the largest inferred spiking response (thus based on the response to the preferred eye). For neurons in retinotopic mapping experiments, we calculated the preferred azimuth and elevation using the same approach as for orientation and direction. Finally, all significantly tuned neurons were grouped in a single volume spanning 60 μm in layer 4 or 360 μm from upper layer 2/3 to layer 5.

## Ipsi-cluster detection

Volumes were split in a set of ipsilateral and contralateral eye preferring neurons by thresholding at an ODI value of 0. For layer 4 volumes, all significantly visually responsive neurons were included, and for layer 2/3 to layer 5 volumes only neurons between 370 and 430 μm in depth were included. Clusters of ipsilateral eye preferring neurons were identified using the 'fast search and find of density peaks' algorithm[25]. In brief, the local density for neuron $i$ ($\rho_i$) was calculated following Eq. 2, where $N$ equals the total number of neurons, $d_{ij}$ is the

Euclidian distance of neurons $i$ and $j$ (µm in cortical space), and $d_c$ is a cutoff-distance set to the fifth percentile of the distribution of all local distances. The function $\chi(x)$ returns 1 when $x$ assumes values below zero, otherwise it returns 0.

$$\rho_i = \sum_{j}^{N} e^{-\left[\frac{d_{ij}\chi(d_{ij}-d_c)}{d_c}\right]^2} \tag{2}$$

Next, the distance of each neuron $i$ to the nearest point of higher local density ($\delta_i$) was determined following Eq. 3.

$$\delta_i = \min_{j:\rho_j>\rho_i}\left(d_{ij}\right) \tag{3}$$

Peaks in the local density of ipsilateral eye preferring neurons, ipsi-cluster centers, were identified as neurons with an $\rho_i$ exceeding 0.2 · $\rho_{max}$, and a $\delta_i$ exceeding 0.2 · $\delta_{max}$.

Typically, in the L4 volumes (one volume per mouse, spanning about 50 percent of their binocular visual cortex area), the algorithm detected one to five ipsi-cluster centers (mean: 2.9 ± 1.0 s.d.; M01, M02, M03, M05, M06 and M08: 3 clusters; M04: 1 cluster; M07: 5 clusters; M09: 2 clusters). However, in some randomized control datasets (see below), the algorithm detected on average more clusters (Shuffle ODI: 3.9 ± 1.3 s.d.; Uniform XY: 5.2 ± 1.2 s.d.; Random clusters: 2.9 ± 1.0 s.d.). In order to increase comparability between the real data and the control datasets, we included only the three clusters with highest $\rho_i$ and $\delta_i$ values in group-wise quantitative analyses. The mean numbers of clusters per L4 volume (i.e., per mouse) in those analyses were therefore for real data 2.7 (±1.0 s.d.), for shuffle ODI data 2.9 (±0.4 s.d.), for uniform XY data 3.0 (±0.1 s.d.) and for random clusters data 2.7 (±0.7 s.d.).

## Randomization procedures

We performed several randomization controls. For the "Shuffled ODI" control we randomized ODI values with respect to neuron identity. For the "uniform XY" control we drew for each neuron new XY coordinates from a uniform distribution ranging from the minimum to maximum of the original set of neuron positions. For the "Random clusters" control we randomly sampled cluster-centers (XY coordinates) excluding regions closer than 200 µm to the field of view border. For this control, we always sampled the same number of random clusters as the number of actual ipsi-clusters that were detected in the field of view. Finally, for the "Swapped ODI" control we iteratively selected the pair of neurons, without replacement, that was spatially separated by a distance closest to a specified reference distance (50 µm, 100 µm,…, 250 µm) and swapped their ODI values. We continued this process until all neurons were paired up and had their ODI values swapped. Thus, every neuron was only swapped once. Because of this restriction, typically five to 20 final pairs were separated by distances deviating more than 50 µm from the specified distance. In all controls, ipsi-cluster centers were identified anew after the randomization procedure.

## Cross correlation maps

Spatial auto- and cross-correlation maps were calculated based on smoothed single-neuron ODI maps, which were constructed as follows. A 'summed ODI map' was calculated by, for each pixel, summing the ODI values of all neurons that had their centroid at that pixel. Next,

a 'coverage map' was constructed, in which the value of each pixel reflected the number of neurons having their centroid at that pixel. Finally, a smoothed single-neuron based ODI map was obtained by dividing the 'summed ODI map' by the 'coverage map' after smoothing the maps with a two-dimensional Gaussian kernel ($\sigma = 16$ µm). Using the pixel-wise Pearson product-moment correlation between two smoothed ODI maps at varying spatial offsets (Eq. 4) we obtained the auto and cross-correlation maps[63,82].

$$r\left(o_y, o_x\right) = \frac{n\sum_y^{N_y}\sum_x^{N_x}M1_{(y,x)}M2_{(y-o_y,x-o_x)} - \sum_y^{N_y}\sum_x^{N_x}M1_{(y,x)}\sum_y^{N_y}\sum_x^{N_x}M2_{(y-o_y,x-o_x)}}{\sqrt{n\sum_y^{N_y}\sum_x^{N_x}M1_{(y,x)}^2 - \left(\sum_y^{N_y}\sum_x^{N_x}M1_{(y,x)}\right)^2}\sqrt{n\sum_y^{N_y}\sum_x^{N_x}M2_{(y-o_y,x-o_x)}^2 - \left(\sum_y^{N_y}\sum_x^{N_x}M2_{(y-o_y,x-o_x)}\right)^2}} \tag{4}$$

The cross-correlation $r$ was calculated at each pixel offset ($o_x$ and $o_y$) individually. $N_y$ and $N_x$ represent the number of pixels along the y and x dimensions in the smoothed ODI maps $M1$ and $M2$, $n$ is the total number of pixels in the map, and $M1_{(y,x)}$ returns the ODI value of map $M1$ at pixel position $(y,x)$. The cross-correlation was only calculated for pixel-offsets at which minimally 20% of the smoothed ODI maps overlapped.

## Relation between feature maps for ocular dominance, azimuth, elevation, and orientation bias

We produced smoothed feature maps ($\sigma = 42$ µm), as described in the previous paragraph, for ODI, preferred azimuth, preferred elevation, preferred orientation, and receptive field width (Fig. 3a, b). Preferred azimuth and elevation were determined as the azimuth or elevation that resulted in the maximum response amplitude in a tuning curve that was averaged over the other feature (elevation or azimuth respectively). Preferred orientation was calculated by fitting a two-peak Gaussian curve[83] to the tuning curve representing the responses to the eight presented drifting grating directions, for the eye that gave the largest response. Receptive field width was measured as the sigma of a single peak Gaussian curve fitted to the tuning curve for azimuth (neuronal responses averaged across elevations). We did not include elevation in the estimation of receptive field width, as elevation was only sampled at three discrete points in visual space (and thus could not be fitted reliably). Regions of feature maps that had fewer than 10 neurons contributing were blanked.

From the azimuth and elevation maps, we calculated the direction and magnitude of the change in preferred azimuth and elevation across the field of view, and used those gradients to compute the CMF[36,84] in mm²/degree². In addition, we used the direction and magnitude of the azimuth and elevation gradient to estimate, for each pixel, the shape of the cortical representation of a square part of the visual field and used the vertices of this shape to calculate a pixel-by-pixel representation of the retinotopic distortion vector $R_s$[37] and a variable representing the relative stretch of the map for azimuth versus the map for elevation ($M_{min}/M_{max}$) across the entire field of view.

From the feature map for azimuth, we algorithmically outlined a region roughly corresponding to binocular V1. We identified the lateral border of V1 from the direction of the gradient for preferred azimuth, as the direction of change in the azimuth map reverses across the higher area boundary. We identified the medial boundary by thresholding the azimuth map at 40 degrees azimuth. In addition, we imposed further boundaries by setting a margin of 42 µm from the map edges inwards.

Geometrically defined ODI clusters were identified in a highpass-filtered ODI map, which was calculated by subtracting two ODI maps, smoothed with kernels that differed in size by a factor of 2.5 (i.e., subtracting a map smoothed with $\sigma = 42$ µm from a map smoothed with $\sigma = 105$ µm; see Supplementary Fig. 12a). The highpass-filtered ODI map was thresholded at the 15$^{th}$ percentile of the highpass ODI

values, and individual continuous regions were identified as ODI clusters if these matched the following criteria: (1) The length of the region (long axis of fitted ellipsoid) was larger than 100 µm; (2) the actual (not highpass) ODI value at the center coordinate was lower than the 25th percentile of actual ODI map values; (3) the center coordinate of the cluster fell within the boundary of V1; and (4) the center coordinate of the cluster fell within an area of the ODI map in which at least 10 cells contributed to the observed ODI map values. In order to similarly detect putative 'contra clusters', we applied the procedure on a map with inverted ODI values (see Supplementary Fig. 12b). However, as these "clusters" did not exhibit any features different from shuffled data, we refer to these as "contralateral eye preferring regions".

We verified that the two methods for detecting ipsi-clusters resulted in similar outcomes by quantifying the number of density-based ipsi-clusters that fell within the outlines of geometrically defined ipsi-clusters. Note that we could not directly calculate the overlap of clusters detected by the two different methods because the density-based method only provided center coordinates for ipsi-clusters. From the 26 density-based ipsi-clusters, 18 fell within the outline of a geo-metrically defined ipsi-cluster, and three were located right outside of the edge of a geometrically defined ipsi-cluster. Two clusters fell out-side the region defined as binocular V1, and three density-based ipsi-clusters were located in regions that exhibited low ODI values, but were not part of geometrically defined ipsi-clusters. Thus, the vast majority of density-based ipsi-clusters were detected in regions that were part of geometrically defined ipsi-clusters.

### Statistics
Data analysis and statistical testing was performed using Python (3.8.12–3.12.3), Numpy (1.20.3–1.26.4), and Scipy (1.7.3–1.13.0). We did not use statistical methods to predetermine the sample size, but used an $n$ similar to that reported in previous publications[14,15]. We excluded data of one Scnn1a-Tg3-Cre transgenic mouse (dataset shown in Supplementary Fig. 7) because the signal-to-noise ratio was low and Suite2P only detected ~300 neurons in the entire L4 volume, which was not sufficient for identifying densities of ipsi-lateral eye preferring neurons. Experimenters were not blinded to experimental conditions. All data are represented as mean (±s.e.m.) unless indicated otherwise. Group-wise differences were tested using non-parametric statistical tests, the Wilcoxon matched-pairs signed-rank (WMPSR) test for two-group matched-samples comparisons and the Kruskal-Wallis test for multi-group comparisons (data were nor-malized per mouse), with alpha set to 0.05. For cross-sections of cross-correlation maps, we used a linear mixed-effects model (mixedlm, Statsmodels 0.14.2) to test for a difference between real and shuffled data across the potentially correlated measurements along the spatial axis of the cross-sections. The interaction of condition (data vs. local shuffle) and measurement (distance from center) indicated which spatial bins were significantly different between conditions with an alpha of 0.01.

### Reporting summary
Further information on research design is available in the Nature Portfolio Reporting Summary linked to this article.

## Data availability
The data generated in this study are publicly available at https://gin.g-node.org/pgoltstein/mouse-od-columns/.

## Code availability
The Python code used for data analysis and production of figures is available on https://github.com/pgoltstein/mouse-od-columns/. Custom-written MATLAB and Python routines used for data collection and data preprocessing are available upon request.

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

## Acknowledgements
We thank Max Sperling, Volker Staiger, Claudia Huber, Frank Voss, and Ursula Weber for technical assistance. This project was funded by the Max Planck Society and the Collaborative Research Center SFB870 (project number A08, reference number 118803580) of the German Research Foundation (DFG) to M.H.

## Author contributions
P.M.G. and M.H. designed the experiments. P.M.G. and D.L. conducted experiments. P.M.G. programmed analysis code and analyzed data. P.M.G., D.L., T.B., and M.H. discussed the data and wrote the manuscript.

## Funding

## Competing interests
The authors declare no competing interests.
