## [Transparent Peer Review file · Nature Communications]

A column-like organization for ocular dominance in mouse visual cortex

Corresponding Author: Dr Pieter Goltstein

Version 0:

Reviewer comments:

Reviewer #1

(Remarks to the Author)

Goltstein et al. conducted volumetric two-photon imaging in the binocular areas of the mouse visual cortex and claimed that cells preferring ipsilateral eye tend to form spatial clusters spanning from the superficial to deep cortical layers. While previous anatomical studies in rats, a closely related rodent species, suggested the presence of ipsilateral eye-specific domains in layer 4 of the binocular visual cortex (ref 21, 22), the present study provides the functional evidence for micro-architectures of ocular dominance in the rodent visual cortex and showed that this spatial architecture extended beyond layer 4.

Columnar functional architecture is one of the most important features of the cerebral cortex. How and why the functional columns are formed in the specific species are important topics in the field of neuroscience. Specifically, ocular dominance columns are well known in carnivores and primates, but has not been well studied in rodents (especially mice). The findings in mice reported in this study provide knowledge for future studies including both experimental and theoretical research.

However, as described below, the results and analyses do not seem sufficient to support the authors' arguments. In addition, a similar type of ocular dominance bias has already been reported in rats, which somewhat weakens the significance of this study.

Major comments:

1) In the ocular dominance columns in higher mammals, both ipsi- and contralateral- preferring neurons are almost perfectly segregated in layer 4. In contrast, in the mouse visual cortex, these neurons are intermingled even at the ipsilateral cluster centers (the average ODI at ipsilateral cluster centers is near zero, as shown in Figure 1). Given this substantial inter-species differences, the use of the term "ocular dominance columns" may be an overstatement. The authors should carefully reconsider the terminology when describing the architecture found in mice. Further, in L81-82: The mean ODI near the cluster center is close to zero, indicating that most neurons responded similarly to both eyes, or approximately half of the nearby neurons responded the ipsilateral eye. Thus the description "most nearby neurons indeed responded to preferentially to the ipsilateral eye." (L81-82) may be somewhat exaggerated. Throughout Discussion, the existence of ocular dominance columns in mice is compared to other species, but it is difficult for me to agree with the existence of ocular dominance columns from the results of this study.

2) To confirm the columnar structure, the authors computed cross-correlation of ODI maps between L4 and other layers and argued the similarity of the ODI maps across layers based on the near-center distribution of peak positions of the cross-correlation maps (Figure 2g). This analysis does not seem sufficient to support the argument. In the cross-correlation maps (Figure 2f), peaks are not clear, and only the axis of the binocular zone (red band) is visible. While cross-correlation analysis in Figure 2f is intended to visualize periodicity of columnar patterns across cortical sheets, the analysis resulted in the visualization of thick vertical stripes with high correlation values, presumably corresponding to the structures of V2 / V1B / V1M. Therefore, the results of this analysis may only indicate the presence of binocular zones, rather than the columnar

architectures of ocular dominance. To show columnar organizations of ocular dominance more directly, the data displayed in Extended data Figure 5b, d, f, h, j may be more informative. The authors should quantitatively analyze the vertical alignment of the similar ocular dominance in the y-z plane of the imaging stacks and present these results in main figures. However, visual inspection of Extended data Figure 5b, d, f, h, j suggests that columnar organization is not always evident.

3) As described in the text, a similar type of ocular dominance bias has already been reported in the rat. I recognize that brain size is different between rat and mouse, and the existence of the ocular dominance bias in mouse cortex is an important finding for considering the mechanisms of columnar structure formation. However, I feel that the impact of this manuscript is somewhat weak.

4) The critical aspect of the analysis that may impact the conclusion is the methods used to detect neuronal ROIs and remove contaminations of neuropil signal. Automated ROI detection methods based on Suite2p sometimes detect non-neuronal ROIs such as dendrites or axons. For example, identifying dendritic ROIs from the same layer 5 pyramidal neuron at multiple imaging planes could lead to the incorrect conclusion that cells with same functional preference are vertically aligned. Furthermore, neuropil signal contaminations may make responses of neighboring neurons appear similar. Although the neuronal ROI detection method used in this study (Suite2p) seems to offer some functions for ROI selection and neuropil signal correction, the authors should clarify how these issues were addressed and present data without these problems.

5) The column structure of ocular dominance in V1 should reflect the anatomical projections from dorsal lateral geniculate nucleus and / or callosal projections from contralateral hemisphere. Addressing this point must increase the impact of this study. ructure.

Minor comments

1) One of my concerns is the trial-to-trial reliability of the ocular dominance index (ODI) in individual neurons and the spatial clustering of ipsi-preferring neurons. Contra- and ipsi-preference were defined by positive and negative ODI values, respectively (L409-410). In addition, stimuli were presented a relatively small number of times (10 and 5 times for L4 and L2/3-5 experiments, respectively. L343). Under these conditions, neurons whose ODIs are around zero could easily change their preference if their responses were unreliable. This should also affect the cluster center detection. Please check whether the main results (the spatial clustering and the columnar structure) are consistent across trials. For example, if the data were split into the first and second half of the trials, are the results consistent between them.?

2) In the analysis of the relationship between ODI and distance to cluster center (e.g., Figure 1e), ODIs were averaged across all responsive neurons. If only the ipsi-preferring neurons were focused, is the ipsilateral bias highest at the center of cluster? How about the trial-to-trial reliability of ocular dominance? Are there any functional differences between inside and outside the cluster? In addition, the functional significance of the cluster of ipsi-preferring neurons should be discussed.

3) In higher mammals, the spatial arrangements of orientation preference maps are related to those of ocular dominance maps. How is the relationship between ODI and preferred direction/orientation in mouse visual cortex? Although a representative result shown in Extended Data Figure 1d suggests that there is no clear relationship between two factors, quantitative analyses and clear descriptions should be informative.

4) L77-80: "If ipsi-clusters merely reflected an overall uneven spatial distribution of neurons, there should be a similar pattern for contra preferring cells, and the average ODI near ipsi-clusters should be relatively similar to its surround." This reasoning is wrong. Since the ipsi-cluster is chosen as the peak of ipsilaterality, the cluster should have a smaller value than its surround, even for the random distribution.

5) L119 (and other parts in legends): What does the WMPSR indicate? Please spell it out. Also describe it in the Method section.

6) L358. Figures 1b and 2a have no maps. Please refer to the correct figure panels.

7) L435-438. "For the "Random clusters" control we drew new XY coordinates for the originally detected ipsi-clusters, also from a uniform distribution, but with the range set 200 μm inwards from the minimum and maximum of the neuron positions." The description is difficult to understand.

8) L438-448. "Swap ODI" "We continued this process until all neurons were paired up and had their ODI values swapped (typically five to 20 final pairs were separated by distances deviating more than 50 μm from the specified distance)" Does it mean that swap was done only once for one specific neuron?

Reviewer #2

(Remarks to the Author)
See attached file

Reviewer #3

(Remarks to the Author)

Goldstein and colleagues address the question to what extent mouse visual cortex exhibits a functional organization described in many other mammals, by focusing on eye preference and ocular dominance columns. Using large-scale calcium imaging in lightly anesthetized mice and drifting grating stimuli to the left or right eye, they identify ocular dominance columns in mouse primary visual cortex. These columns are characterized by an input preference from one eye over the other and stretch across cortical layers.

The study addresses an important problem in neuroscience, which is whether cortical columns are a general organizational principle of the mammalian cortex. The paper is well-written, and the data is presented clearly. The authors include many additional data and controls in the supplement and overall, the claims are supported by the data. However, I have two concerns:

First, in line 189 the authors write that “In several mice, pixelwise ODI maps across different cortical depths showed a clear similarity in the overall patterns of ipsilateral and contralateral eye dominated regions”. They should specify what “in several mice” means. Does that mean that the consistency across layers was only statistically significant for some mice? They discuss the variability across animals later, but it would be important to be more explicit in the Results section. In addition, some of the variability across mice is not mentioned and discussed at all. Fig. 1g shows that some mice have ipsi-clusters with a preference for the ipsilateral eye while the ipsi-clusters of other mice only have a weaker preference for the contralateral eye. This should be mentioned and discussed with respect to what is known about inter-animal variability with respect to ocular dominance columns. As much raw data for each animal should be shown as possible including in main figures.

Second, the term “ocular dominance column” is not clearly defined in the paper. For example, would the existence of two ipsi-clusters be sufficient to call the observed functional organization a column? Or does it require a repetitive motif, as observed in other species with ocular dominance and orientation columns? The study still addresses an important question, but the observed organization looks rather like ocular dominance domains than columns. At the very least, the authors should define the term and discuss these problems.

Version 1:

Reviewer comments:

Reviewer #1

(Remarks to the Author)

My initial concern during the 1st round was that the results and analyses presented were insufficient to support the claim that the cluster of ipsi-preferring neurons constitutes a columnar structure. The authors have sincerely addressed this concern by providing additional analyses and discussions that have led to a more accurate understanding of the ocular dominance (OD) structure in mice. In higher mammals, OD columns consist of completely segregated groups of cells that respond to either ipsi or contra inputs, with these ipsi- or contra-preferring groups of cells extending across cortical layers. However, the OD structure discussed in this paper, specific to mice, differs from those in higher mammals. In mice, clusters of cells that respond more strongly to ipsilateral inputs exist at some locations within the binocular region of V1 and are mixed with contra-preferring neurons and binocular neurons. So I still have concerns about the terminology about the “columnar” organization of ocular dominance. In addition, there are some unclear points in the additional analyses. To make this paper more accurate, I believe the authors need to revisit these points once again.

Major comments

1) My main concern is with the terminology related to “columns”. In the revised manuscript, the authors define the vertically extending clusters as columns. However, this term naturally reminds many readers of functional clusters that extend consistently in shape across layers, as observed in cats and monkeys. The data shown in the manuscript (e.g., Extended data Figures 6, 7, 9) indicate that the cluster shapes are only weakly similar across planes. For example, in the new result shown in Extended data Figure 9, although the peaks are significantly different from the shuffled control, the differences are very small. These suggests that the cross-correlation peaks are mostly due to the global structure, while the contributions of the local structures are very small. Further, as seen in Extended data Figure 6, the ODI map patterns are only weakly preserved between L4 and other layers. This type of functional organization is far from what the OD columns usually observed in other species. Therefore, I recommend that the authors use other terms and accurately describe the results without exaggeration throughout the manuscript including the title and abstract.

2) Regarding the cross-correlation map in Figure 2f, the peaks in the cross-correlation maps are still unclear. Extended data Figure 9 is important to show quantitatively that the peaks of the cross-correlation maps had very small differences from the shuffled control and should be presented in a main figure.

In the statistical analyses of Extended data Figure 9, the Kruskal-Wallis test gives a very small p-value of 10^{-36} , even though there is little difference from the shuffled control, suggesting that perhaps the authors may be using the degrees of freedom incorrectly. I speculate that the Wilcoxon matched-pairs signed rank test was applied at each distance point and repeated many times and the multiple comparison problem is not properly addressed.

3) The authors have demonstrated that neurons responding to ipsilateral inputs form clusters. However, it remains unclear whether these neurons are ipsi-preferring neurons or binocularly responsive neurons. As the authors noted, the average ODI (Ocular Dominance Index) is close to zero even near the center of the clusters (Fig. 1g). Based on the current data, it is

unclear whether this is due to the mixing of ipsi- and contra-preferring cells or simply because the clusters are composed of binocularly responsive neurons. To clarify this, the authors should present the ODI distribution of individual neurons from the ipsi-preferring clusters. If there are many cells with ODIs below some threshold (e.g. -0.3) in the clusters, they would be considered ipsi-preferring clusters. On the other hand, if neurons with ODIs close to zero predominate, they would be considered binocularly responsive clusters. If the latter is the case, I believe that the paper would be more accurate if it discussed binocularly responsive clusters rather than ipsi-preferred clusters.

In the additional analysis to determine the percentage of ipsi-preferring neurons (Extended Data Figure 4), the authors cut off the ipsi-preferring neurons at $ODI = 0$, and if they are even slightly negative, they call them ipsi-preferring neurons. However, for example, neurons with $ODI = -0.01$ cannot be called ipsi-preferring neurons, but rather binocular neurons. The authors should define ipsi-preferring neurons when the response of ipsi is statistically significantly greater than that of contra, or when some threshold (e.g. -0.3) is set on the ODI.

Minor points

L653-655: According to L653, 1-5 clusters were detected in each volume. On the other hand, according to L655, the top three clusters were used in the "group-wise quantitative analyses". Does this mean that in each mouse, multiple volumes were recorded, at least three clusters were detected, and the top three clusters among them were averaged in each mouse and the averaged values were used for the "group-wise quantitative analyses" with a sample number of 9mice? How many clusters were detected by the method (ipsi cluster detection in L633-655) in each mouse? Describe these points clearly. Also specify in which analyses or figures "the group-wise analyses" are used.

A new method was used to geometrically identify ipsi- and contra-clusters. According to the L654-655 and the Table 1, the number of clusters is different between those included in the "group-wise quantitative analyses (L654)" and those geometrically detected by the new method (Table 1). How many clusters overlap between them?

The explanations of the new method in the Methods (L721-733) seem to be inconsistent with those in the legend of Extended data Figure 10 (L959-977). For example, the thresholds for the high-pass filtered maps were different (15th percentile in L724 vs. 25th percentile in L962). Please describe the explanation consistently in these two parts.

L722: In "by subtracting two ODI maps", please specify which map was subtracted from which map (as in L962).

L263: Please add explanation of term, "V1b" on first use.

L288: typo. 'In order to asses...' should be corrected to 'In order to assess...!.'

In the middle and bottom panels of Extended data Figure 7a and b, please indicate the reference (or origin) points to which the ipsi-cluster centers are aligned. In addition, it should be helpful to draw rough boundaries between layers.

In the Scnn1a-cre mice, data were obtained from the right hemisphere. From which hemisphere were the other data recorded? The horizontal flip of the maps between Extended data Figure 5c and other related maps may confuse the readers unfamiliar with the mouse visual cortex.

Reviewer #2

(Remarks to the Author)

The authors have fully addressed all my comments. Their careful analysis and detailed functional measurements of mouse visual cortex (including the extensive supplementary data) provide new important constraints to developmental models of visual cortical topography.

First, we like to thank you for the constructive comments and criticisms on our manuscript “Ocular dominance columns in mouse visual cortex”.

In line with your suggestions, we have now updated the manuscript with several new analyses describing, in more detail, the properties and reproducibility of the ocular dominance clusters in layer 4 and their vertical alignment in columnar-like structures. Additionally, we have added an in-depth investigation of the potential interaction of the functional organization for ocular dominance with the retinotopic map (and a map for local bias in preferred orientation). Finally, we have refined and amended our use of terminology for describing the findings, paying specific attention to how we define, and refer to, the observation of vertically extending clusters. Detailed responses to individual comments are listed point-by-point below.

We believe that with these changes, our revised manuscript provides significant new insights into the functional organization of ocular dominance in mice, going beyond what studying c-Fos expression in rodents has been able to reveal by precisely describing how key functional properties of neurons (i.e. eye preference, retinotopic preference, orientation preference) interact within the context of identified feature maps. We hope that the updated manuscript addresses all raised questions and concerns, and we are looking forward to your response.

Best regards,

Pieter Goltstein, David Laubender, Tobias Bonhoeffer and Mark Hübener

Reviewer #1 (Remarks to the Author):

Goltstein et al. conducted volumetric two-photon imaging in the binocular areas of the mouse visual cortex and claimed that cells preferring ipsilateral eye tend to form spatial clusters spanning from the superficial to deep cortical layers. While previous anatomical studies in rats, a closely related rodent species, suggested the presence of ipsilateral eye-specific domains in layer 4 of the binocular visual cortex (ref 21, 22), the present study provides the functional evidence for micro-architectures of ocular dominance in the rodent visual cortex and showed that this spatial architecture extended beyond layer 4.

Columnar functional architecture is one of the most important features of the cerebral cortex. How and why the functional columns are formed in the specific species are important topics in the field of neuroscience. Specifically, ocular dominance columns are well known in carnivores and primates, but has not been well studied in rodents (especially mice). The findings in mice reported in this study provide knowledge for future studies including both experimental and theoretical research.

However, as described below, the results and analyses do not seem sufficient to support the authors' arguments. In addition, a similar type of ocular dominance bias has already been reported in rats, which somewhat weakens the significance of this study.

Major comments:

1) In the ocular dominance columns in higher mammals, both ipsi- and contralateral- preferring neurons are almost perfectly segregated in layer 4. In contrast, in the mouse visual cortex, these neurons are intermingled even at the ipsilateral cluster centers (the average ODI at ipsilateral cluster centers is near zero, as shown in Figure 1). Given this substantial inter-species differences, the use of the term "ocular dominance columns" may be an overstatement. The authors should carefully reconsider the terminology when describing the architecture found in mice. Further, in L81-82: The mean ODI near the cluster center is close to zero, indicating that most neurons responded similarly to both eyes, or approximately half of the nearby neurons responded the ipsilateral eye. Thus the description "most nearby neurons indeed responded to preferentially to the ipsilateral eye. "(L81-82) may be somewhat exaggerated. Throughout Discussion, the existence of ocular dominance columns in mice is compared to other species, but it is difficult for me to agree with the existence of ocular dominance columns from the results of this study.

We agree with the reviewer that the vertical grouping of eye preference as we observe it in the mouse is not a one-to-one mirror image of what has been found previously in higher mammals (e.g. zebra-crossing like patterns in macaque monkeys). While the mean ODI of -0.07 (as reported on line 91) does indicate that the ipsilateral eye is on average dominant in ipsi-clusters, the contralateral eye indeed also still provides a significant drive. We now better characterize the response properties of neurons in ipsi-clusters, reporting 1) the fraction of ipsi-preferring neurons, 2) the response amplitude and 3) ODI of ipsilateral and contralateral eye preferring neurons separately (Extended data Fig. 4a-f).

In brief, the new analyses indicate that the ipsilateral eye provides a strong input to neurons in ipsi-clusters, that slightly more than 50% of neurons prefer the ipsilateral eye, but that indeed also the contralateral eye still provides a significant input. In line with this, we have updated the main text in the results section to better reflect the assessment on lines 91-94: "*However, the mean ODI near ipsi-clusters was just below zero (-0.07 ± 0.15 s.d.), indicating that many nearby neurons indeed responded preferentially to the ipsilateral eye. Furthermore, ODI increased with distance from the cluster center, showing that neurons outside the ipsi-clusters generally preferred the contralateral eye (Fig. 1e, g).*"

Regarding the question of the existence of ocular dominance columns: Our data show that the vertically extending functional clusters of ipsi-preferring neurons in the mouse visual cortex are clearly beyond what would be expected from random processes. The description of a vertically aligned

arrangement of neurons with similar eye preference matches the definition of a columnar system, in which neurons would lie in narrow vertical columns or cylinders and would be activated by similar class of stimuli (paraphrased from Mountcastle, *J Neurophysiol*, 1957).

In our manuscript we, in general, refer to the observed vertically extending clusters as columns for ocular dominance, or in short, ocular dominance columns. However, we do acknowledge that the exact pattern of ipsi-clusters in mice, which shows regions of high binocularity, does not directly match the pattern of alternating ipsilateral and contralateral eye preferring neurons as reported in higher mammals. Thus, in our manuscript, we provide two levels of descriptive terminology. We refer to the groups of ipsilateral eye preferring neurons as ipsi-clusters, and describe the observation of vertical alignment of ipsi-clusters as columns for ocular dominance, in line with the terminology used in earlier studies on rodent visual cortex (e.g. Laing et al., *Cereb Cortex*, 2015; Andelin et al., *J Comp Neurol*, 2020; Zhou et al., *Cereb Cortex*, 2023).

Finally, we have added a paragraph in the discussion (on lines 397-418) that addresses differences between the columns for ocular dominance as observed in higher mammals, and our observations in mice, providing context to the question of whether ocular dominance columns in mice and higher mammals are fundamentally the same type of structures.

2) To confirm the columnar structure, the authors computed cross-correlation of ODI maps between L4 and other layers and argued the similarity of the ODI maps across layers based on the near-center distribution of peak positions of the cross-correlation maps (Figure 2g). This analysis does not seem sufficient to support the argument. In the cross-correlation maps (Figure 2f), peaks are not clear, and only the axis of the binocular zone (red band) is visible. While cross-correlation analysis in Figure 2f is intended to visualize periodicity of columnar patterns across cortical sheets, the analysis resulted in the visualization of thick vertical stripes with high correlation values, presumably corresponding to the structures of V2 / V1B / V1M. Therefore, the results of this analysis may only indicate the presence of binocular zones, rather than the columnar architectures of ocular dominance.

To show columnar organizations of ocular dominance more directly, the data displayed in Extended data Figure 5b, d, f, h, j may be more informative. The authors should quantitatively analyze the vertical alignment of the similar ocular dominance in the y-z plane of the imaging stacks and present these results in main figures. However, visual inspection of Extended data Figure 5b, d, f, h, j suggests that columnar organization is not always evident.

The cross-correlation maps, as shown in e.g. Fig. 2f, show two effects. Firstly, indeed as the reviewer indicates, there is a broad vertical band of positive cross-correlations, which reflects similarity across layers of the broader structure of binocular V1. Secondly, in the center of the cross-correlation maps, there are narrow peaks, which indicate that precise vertical alignment at a smaller scale results in even higher cross-correlations across layers. In order to visualize the narrow peak better, we now provide cross-sections of the cross-correlation maps (Extended data Fig. 9). In this figure, we show the cross-correlation for the actual data in direct comparison to local shuffle control data (i.e. 'swapped ODI': ODI values are swapped between pairs of neurons at ~200 μm distance). This control shuffles local/precise structure in ODI maps, while keeping the overall structure of the binocular zone intact (see Extended data Fig. 2e,f for an illustration of the method).

In Extended data Fig. 9, the effect of the binocular zone on the cross-correlation can be seen in both the actual data as well as the local shuffle control. In panel 9a, the cross-correlation along the medial-lateral axis drops off to negative values at distances of 400-500 μm , thus indicating broadly consistent ODI values up to distances approximately consistent with the width of the binocular zone along the medial-lateral axis. In panel 9b, which is the cross-section along the anterior-posterior axis, there is little drop-off, in accordance with the binocular zone being more elongated along this axis (see e.g. Extended data Fig. 2a for a larger imaging field of view showing the full outline of binocular V1). In addition, the effect of alignment of more fine-grained structures in ODI maps across layers (i.e. columns for ocular dominance) can be seen by comparing the actual data with the local shuffle control. In panel 9a

and panel 9b there is a clear narrow peak in the cross-correlation between L4 and L2/3, and between L4 and L5, relative to the local shuffle control. This highlights that there is a pattern within the ODI maps with a resolution much finer than the overall binocular zone that aligns across cortical layers.

At the request of the reviewer, we have added the suggested analysis that displays the average vertical arrangement of ODI from a side view perspective. For each ipsi-cluster we created side-view ODI images along the anterior-posterior axis and along the medial-lateral axis. These side-views were then averaged such that the X or Y center coordinate of the ipsi-clusters aligned with the center of the figure panel. In these side-views it can be seen that ocular dominance columns extend beyond the narrow depth-range of cortical layer 4 (Extended data Fig. 7a,b). We present these ‘average vertical organization’ figure panels as extended data figures (rather than main figures). We illustrate the vertical structure of ocular dominance, and the correlation of patterns of ipsi-clusters across cortical layers, in the main figures preferentially using examples from individual mice, because these better account for the high variability in the shape of these three-dimensional structures (individual examples can be found in Fig 2a, bottom, and Extended data Fig. 6b, d, f, h, j).

Finally, the reviewer commented that the extent of vertical organization is not always strongly evident, i.e. there is variability across animals. While we already referred to this in the original manuscript, we now explicitly indicate the observed variability across animals in the results section (lines 243-245) and refer to potential sources of this variability in the discussion section (lines 410-416 and 470-476).

3) As described in the text, a similar type of ocular dominance bias has already been reported in the rat. I recognize that brain size is different between rat and mouse, and the existence of the ocular dominance bias in mouse cortex is an important finding for considering the mechanisms of columnar structure formation. However, I feel that the impact of this manuscript is somewhat weak.

We are glad that the reviewer thinks that our finding is important for understanding the mechanisms of column formation. The reviewer is correct in stating that earlier studies indeed have found clustered immediate early gene (IEG) expression and patches of thalamocortical and callosal axons in binocular V1 of the rat (e.g. Laing et al., *Cereb Cortex*, 2015; Andelin et al., *J Comp Neurol*, 2020; Zhou et al., *Cereb Cortex*, 2023). However, while the most recent of these studies (Zhou et al., 2023) does even suggest that a similar pattern is likely present in the mouse, the data is, according to the authors themselves, “... not sufficiently clear to refer as “ODCs” ... ” (ocular dominance columns). Moreover, IEG expression does not always precisely indicate neuronal activity (see for instance Lee et al., *PNAS*, 2021 and Mahringer et al., *Peer Comm Journ*, 2022), and the impact of thalamocortical and callosal anatomical segregation on functional response properties in visual cortex can vary in strength (Ramachandra et al., *Nat Commun*, 2020).

Thus, we think that our manuscript goes substantially beyond these earlier studies by addressing several unanswered questions regarding the functional organization of the visual cortex. (1) Can a region of cortex that is as small as the binocular region of mouse V1 harbor a columnar organization, e.g. for ocular dominance? (2) How do neurons encode visual information within the cortical micro-organization of ocular dominance columns *in vivo*? (3) How does the spatial arrangement of ocular dominance columns relate to other feature maps such as the retinotopic map?

Our study now conclusively shows that during *in vivo* visual processing there are spatially defined clusters of ipsilateral eye preferring cells that align across cortical layers, forming a columnar organization for ocular dominance in mouse primary visual cortex. Moreover, we provide insight into several *in vivo* functional properties of visual cortex neurons in relation to the observed columnar organization. These properties, like response amplitude, neuronal tuning for eye preference and stimulus orientation, as well as the neurons’ receptive field position cannot be determined using IEG experiments. We provide in-depth analyses of the interactions of these properties (Figs. 1 and 2, and Extended data Figs. 3 and 4), as well as their derived functional maps (Fig. 3 and Table 1), with the pattern of ipsi-clusters. Moreover, we show that the observed maps are stable across days and weeks (Extended data Fig.

10c) and we precisely quantify the properties of the cortical map for retinotopy to assess its relationship with the map for ocular dominance in mouse binocular V1 (Fig. 3 and Extended data Fig. 10a,b).

Thus, our study goes beyond the mere finding of ocular dominance columns by providing detailed insights into the functional micro-organization of ocular dominance and retinotopy in mouse visual cortex that can be directly linked to predictions by models for cortical columnar organization (such as in Najafian et al., Nat Commun, 2022). We have now made this link more explicit by adding several analyses suggested by reviewer #2 (see Fig. 3, Extended data Fig. 10; Results, lines 274-369; Discussion, lines 435-446).

4) The critical aspect of the analysis that may impact the conclusion is the methods used to detect neuronal ROIs and remove contaminations of neuropil signal. Automated ROI detection methods based on Suite2p sometimes detect non-neuronal ROIs such as dendrites or axons. For example, identifying dendritic ROIs from the same layer 5 pyramidal neuron at multiple imaging planes could lead to the incorrect conclusion that cells with same functional preference are vertically aligned. Furthermore, neuropil signal contaminations may make responses of neighboring neurons appear similar. Although the neuronal ROI detection method used in this study (Suite2p) seems to offer some functions for ROI selection and neuropil signal correction, the authors should clarify how these issues were addressed and present data without these problems.

Indeed, it is important to rule out that potential artifacts resulting from neuropil contamination affect the finding. Below, we provide new control analyses and argumentation showing that the alternate hypothesis of ipsi-clusters / ocular dominance columns originating from non-cell specific fluorescence signals is unlikely.

First, we considered whether ipsi-clusters in L4 or upper layers could, in part, reflect clustering of similarly tuned dendrites of cortical L5 cells (see Kondo et al., Nat Commun, 2016).

(1) The clustering is most evident in layer 4 (Fig. 2), which argues against the L4 clusters being signal inherited from clustered dendrites originating in layer 5.

(2) If ipsi-clusters in layer 4 would reflect clustered dendritic signals from layer 5 cells, the clusters should not be visible when layer 5 cells do not express the calcium indicator. This is the case for the layer 4 imaging data acquired in *scn1a* mice, because this mouse line restricts expression of the calcium indicator to layer 4. Thus, in this mouse line there are no axons nor dendrites from layer 5, nor from layer 2/3, in the layer 4 imaging volume, but still ipsi-clusters were observed (Extended data Fig. 5; lines 184-192).

(3) We have investigated the radius, area and aspect ratio of the ROI footprints. If dendritic ROIs from layer 5 neurons were directly the source of neurons with clustered eye preference, it would be expected that there will be a substantial set of ROI footprints with either small radii and areas (for dendrites that run orthogonal to the imaging plane) or with aspect ratios close to one, indicating a round shape (while for dendrites that run within the imaging plane, shapes likely appear very elongated and would result in larger aspect ratios). However, as can be seen in 'Reviewer-Figure 1' below, the distributions of ROI shape quantifications are homogenous and similar across cortical layers. We have added this finding to the methods (lines 593-595).

Next, we investigated whether differences in local neuropil contamination could explain the presence of ipsi-clusters:

(1) In our dataset, fluorescence traces were corrected for neuropil contamination using the standardized Suite2p analysis workflow. In Suite2p the neuropil signal is estimated locally from a region surrounding each ROI that is devoid of cells, and 70% of the local neuropil signal is subtracted from the ROI trace before spike inference. We have updated our results and methods sections on lines 170-174 and 576-593 to describe how the neuropil signal was subtracted.

(2) Ipsi-clusters do not express as regions in which neurons are homogenous in their feature tuning (which would be expected from local neuropil signals), but rather contain a mixed set of cells, which have diverse ODI's (ranging from ipsi- to contra-preferring).

(3) If ipsi-clusters would lie in cortical regions with high neuropil contamination, we would also expect clustering of other tuning features (such as preferred orientation). However, we do not observe clusters of orientation tuned neurons to align with clusters of ipsi-tuned neurons, nor with regions of more contralateral eye preferring neurons (see Fig. 3a,d for example). Neither did we observe clusters of cells with the same retinotopic preference (which would result in distortions in the retinotopic map; Fig. 3) aligning with ipsi-clusters. Thus, this argues against localized differences in the strength of neuropil contamination leading to clusters of similarly tuned cells.

(4) Finally, we have reanalyzed the L4 data using the CaImAn (Giovannucci et al., eLife, 2019) preprocessing and ROI extraction tool. This tool uses a fundamentally different method to demix, denoise and extract ROI signals from background and neighboring neurons. These data show essentially the same clusters (see Extended data Fig. 4g-k and lines 174-182 and 600-608).

In summary, these control analyses all indicate that ipsi-clusters reflect groupings of neurons with similar eye preference, and they make it unlikely that the patches are the result of artifacts relating to neuropil signals (either as non-specific fluorescence background, or as specific dendritic signals).

Reviewer Figure 1: Shape quantification of ROI footprints across cortical layers.

a, Distribution of ROI footprint radii across cortical depths (for depth color-legend, see bars in **b**). Note that several distribution curves are not visible as they are very similar to the foreground curve. **b**, Mean radius for each depth range. **c,d**, As **a,b**, but for the area of the ROI footprints (expressed in number of pixels). **e,f**, As **a,b**, but for the aspect ratio of the ROI footprints.

5) The column structure of ocular dominance in V1 should reflect the anatomical projections from dorsal lateral geniculate nucleus and / or callosal projections from contralateral hemisphere. Addressing this point must increase the impact of this study. ructure.

We agree with the reviewer that this would be interesting to investigate, as it would strengthen the link to the alignment of thalamocortical and callosal projections with ocular dominance columns observed in the rat visual cortex (Laing et al., *Cereb Cortex*, 2015; Andelin et al., *J Comp Neurol*, 2020; Lu et al., *Vis Neurosci*, 2021; Olavarria et al., *J Comp Neurol*, 2021; Zhou et al., *Cereb Cortex*, 2023). While the main goal of our study was to investigate the functional micro-organization of neuronal response properties in the visual cortex, we did make an attempt to investigate the functional properties of thalamocortical axons in relation to the ipsi-clusters.

We performed a series of pilot experiments that were designed to image calcium activity of axons from LGN in L4 (using jGCaMP7b), while in the same animals imaging activity of L4 neurons using the red calcium indicator jRGECO1a. Unfortunately, it turned out that it was not trivial to assess these interactions in the same animal. We were unable to obtain signals from sufficient numbers of LGN axons in mice that also expressed the jRGECO. In our limited data set, we only had recordings with 3-4 LGN axons in a small cortical volume, which does not suffice to make statistically meaningful statements on whether or not specific LGN axons align with the L4 ipsi-clusters. We did not expand our pilot approach to callosal projections. Thus, we are not able to provide more experimental insight into the potential colocalization of ipsi-clusters with patches of similarly tuned LGN or callosal axons.

We now refer in the discussion to the potential role for clustered callosal (lines 405-410) and thalamocortical (lines 435-446) axonal projections playing a role in establishing ocular dominance columns in rodents. However, as the local demixing of LGN axons associated with ocular dominance columns likely results in distortions in local retinotopy (Najafian et al., *Nat Commun*, 2022), the fact that we did not observe such effects makes it possible that other mechanisms than axon segregation contribute to the formation of ipsi-clusters. We believe that a thorough and systematic experimental approach into the spatial distribution and functional properties of afferents in mouse visual cortex might answer these questions in the future, but that such experiments fall beyond the scope of our study, which was designed to provide a detailed account of the functional micro-organization of columns for ocular dominance in mice.

Minor comments

1) One of my concerns is the trial-to-trial reliability of the ocular dominance index (ODI) in individual neurons and the spatial clustering of ipsi-preferring neurons. Contra- and ipsi-preference were defined by positive and negative ODI values, respectively (L409-410). In addition, stimuli were presented a relatively small number of times (10 and 5 times for L4 and L2/3-5 experiments, respectively, L343). Under these conditions, neurons whose ODIs are around zero could easily change their preference if their responses were unreliable. This should also affect the cluster center detection. Please check whether the main results (the spatial clustering and the columnar structure) are consistent across trials. For example, if the data were split into the first and second half of the trials, are the results consistent between them.?

The measurement of the ocular dominance index is consistent across trial-blocks, and the ipsi-clusters can be identified based on single trial-block data. We present these novel analyses in Extended data Fig. 3a-h and lines 137-139.

2) In the analysis of the relationship between ODI and distance to cluster center (e.g., Figure 1e), ODIs were averaged across all responsive neurons. If only the ipsi-preferring neurons were focused, is the ipsilateral bias highest at the center of cluster? How about the trial-to-trial reliability of ocular dominance? Are there any functional differences between inside and outside the cluster? In addition, the functional significance of the cluster of ipsi-preferring neurons should be discussed.

We calculated the ODI as a function of distance to the cluster center separately for ipsilateral eye preferring neurons as well as for contralateral eye preferring neurons. This shows that in general ipsilateral eye preferring neurons are similarly ipsi-selective (only slightly more so near cluster centers as

compared to further away). Similarly, contralateral eye preferring neurons are slightly more ipsi-selective near ipsi-cluster centers. These data are shown in Extended data Fig. 4f and lines 164-166. In addition, we calculated the trial-to-trial variability of the ODI value as a function of distance to ipsi-cluster centers. This variability, expressed as the standard deviation of single-trial calculated ODI values for each neuron, was not different near ipsi clusters as compared to further away (see Extended data Fig. 3i). Finally, in the discussion, we broadly discuss the potential “functional significance of the clustering of ipsi-preferring neurons”, e.g. in supporting the calculation of binocular disparity (lines 453-482).

3) In higher mammals, the spatial arrangements of orientation preference maps are related to those of ocular dominance maps. How is the relationship between ODI and preferred direction/orientation in mouse visual cortex? Although a representative result shown in Extended Data Figure 1d suggests that there is no clear relationship between two factors, quantitative analyses and clear descriptions should be informative.

To address this point, we have used vector averaging to construct “preferred orientation bias” maps. The orientation bias was generally very small (resultant vector length in binocular V1 was approximately 0.12, which was hardly larger than that of randomized maps which have a resultant length of 0.10) and likely reflected local correlations in preferred orientation as observed before (Ringach et al., Nat Commun, 2016; Kondo et al., Nat Commun, 2016). Moreover, we found no spatial relationships between the “preferred orientation bias” map and the map for ODI values, as has been observed in cat and monkey, which we report on lines 356-365 and show in Fig. 3a, b and i.

4) L77-80: “If ipsi-clusters merely reflected an overall uneven spatial distribution of neurons, there should be a similar pattern for contra preferring cells, and the average ODI near ipsi-clusters should be relatively similar to its surround.” This reasoning is wrong. Since the ipsi-cluster is chosen as the peak of ipsilaterality, the cluster should have a smaller value than its surround, even for the random distribution.

Indeed, the ipsi-clusters are detected as peaks in the local density of ipsilateral eye preferring neurons.

There are three potential factors that could contribute to the occurrence and location of these peaks in local density of ipsilateral eye preferring neurons:

(1) There might be areas with increased overall neuron density, leading to peaks that are similar for ipsilateral and contralateral preferring neurons, resulting in the detection of false-positive ipsi-clusters (see “Reviewer Figure 2a”, below). However, because the densities of ipsilateral and contralateral eye preferring neurons are very similar in this case, the mean ODI near peaks in local density would not be different from other regions. Thus, the mean ODI near ipsi-clusters can be used to identify false-positive ipsi-clusters that originate from overall fluctuations in local density.

(2) In addition to the first factor, there can be random fluctuations in the individual densities of ipsilateral and contralateral eye preferring neurons, thus spatially offsetting peaks in ipsilateral and contralateral eye preferring neurons (see “Reviewer Figure 2b”). This would result in the detection of false-positive ipsi-clusters in areas with slight increases in the number of ipsilateral eye preferring neurons. In these areas, we would therefore expect slight reductions in local ODI values. However, such random fluctuations will also be present in shuffled data, thus comparing data to shuffle controls can be used to identify the presence of false-positive ipsi-clusters from random fluctuations.

(3) There could be regions in which ipsilateral eye preferring neurons are much more densely packed than random chance (see “Reviewer Figure 2c”), which would be ‘true’ ipsi-clusters. Here, the local fraction of ipsilateral eye preferring neurons will be higher than the surrounding area by an amount that exceeds chance-level, and therefore the ODI in/near these regions will be lower than the surrounding area in comparison to shuffled data (as can be seen in e.g. Fig. 1g-i).

Thus, the reasoning that false-positive ipsi-clusters will have ODI values that are relatively similar to the surrounding area is supported by these simulations. In addition, the global and local randomizations (shuffle controls) show that, yes, ipsi-clusters can be detected from randomly occurring distributions, but that these clusters do not show the significantly lower ODI values (Fig. 1g-i), nor the strong increases in the fraction of ipsilateral eye preferring neurons (Extended data Fig. 4a-c) as we observe in the real data.

Reviewer Figure 2: Simulation of ipsi-cluster detection.

a, A simulation with three regions having an equal, but high, probability of sampling ipsilateral and contralateral eye preferring neurons (left). The random sample contains similar local densities for ipsilateral and contralateral eye preferring neurons (middle panels). The mean ODI near detected ipsi-clusters (black circles in ipsi-density plot) shows virtually no difference as a function of distance to cluster centers. **b**, As **a**, but for non-discrete density distributions, leading to random mismatches in local density of ipsilateral and contralateral eye preferring neurons. **c**, As **a**, but for a simulation in which only ipsilateral eye preferring neurons show regions of increased local density.

5) L119 (and other parts in legends): What does the WMPSR indicate? Please spell it out. Also describe it in the Method section.

Apologies for the omission. This stands for Wilcoxon matched-pairs signed-rank test, which is a non-parametric two-sample within-subject test. We have updated the statistics section to describe the precise tests used (see lines 744-746).

6) L358. Figures 1b and 2a have no maps. Please refer to the correct figure panels.

Thank you for spotting this, we have corrected the figure references.

7) L435-438. “For the “Random clusters” control we drew new XY coordinates for the originally detected ipsi-clusters, also from a uniform distribution, but with the range set 200 μm inwards from the minimum and maximum of the neuron positions.” The description is difficult to understand.

We have rephrased this such that it better explains the procedure. It reads now: “*For the “Random clusters” control we randomly sampled cluster-centers (XY coordinates) excluding regions closer than 200 μm to the field of view border. For this control, we always sampled the same number of random clusters as the number of actual ipsi-clusters that were detected in the field of view.*” (lines 661-664).

8) L438-448. “Swap ODI” “We continued this process until all neurons were paired up and had their ODI values swapped (typically five to 20 final pairs were separated by distances deviating more than 50 μm from the specified distance)” Does it mean that swap was done only once for one specific neuron?

Yes, each neuron was only swapped once. We have updated the description of the procedure such that it clearly states this: “*We continued this process until all neurons were paired up and had their ODI values swapped. Thus, every neuron was only swapped once. Because of this restriction, typically five to 20 final pairs were separated by distances deviating more than 50 μm from the specified distance.*” (lines 667-670).

Reviewer #2 (Remarks to the Author):

This paper demonstrates subtle but significant neuronal clustering for ocular dominance in the mouse visual cortex. This finding is important because the mouse has become a popular model to study visual cortical circuitry and all studies are currently designed under the assumption that neurons dominated by the ipsilateral eye are randomly interspersed with the more common neurons dominated by the contralateral eye. Ocular dominance columns are thought to be only present in animals with larger binocular fields such as carnivores and primates and some types of rats, but not mice. The finding that small ocular dominance domains can be found in the tiny binocular cortex of a mouse provides new important constraints to developmental models of cortical topography.

The paper is well written, the main results are well documented, the conclusions are solid, and the authors have performed several important controls to clearly demonstrate that the minute ocular dominance clusters are not the result of random (salt-and-pepper) cortical topography. My main comment is that the authors do not seem to be aware of recent work proposing a mechanism of ocular dominance segregation based on the density of thalamic afferents per visual point. According to this mechanism, ocular dominance columns emerge when the number of afferents with overlapping receptive fields (and the iso-retinotopic cortical area accommodating the afferents) are large enough to allow the segregation of afferents and cortical neurons by eye input. Importantly, this mechanism predicts a close relationship between ocular dominance segregation and the retinotopy gradient, which has been demonstrated in humans, macaques and cats (Blasdel and Campbell, 2001; Najafian et al., 2019). The question is whether such relationship also exists in mice. The data from the authors may allow to address this question and it would be a missed opportunity not to do it by performing some additional analysis that should not be too time consuming. More details below.

Major comments

1) Ocular dominance segregation is closely related to cortical retinotopy in several species of mammals. Evidence supporting this relation can be traced back to the work of Hubel and Wiesel demonstrating that ocular dominance stripes run orthogonal to the cortical border between areas V1 and V2 in macaques (Hubel and Wiesel, 1977) and the work of Campbell and Blasdel demonstrating that cortical retinotopy changes slower along than across the V1/V2 border (Blasdel and Campbell, 2001). More recently, a relation between the retinotopic gradient and ocular dominance segregation has been demonstrated for a large portion of area V1 in humans, macaques and cats (Najafian et al., 2019). This close relation between retinotopy and ocular dominance segregation supports the notion that thalamic afferents segregate by eye input in visual cortex when the cortical area representing the same retinotopy is large enough to accommodate neuronal clusters with different eye dominance (Najafian et al., 2019). The same mechanism has been also proposed to explain cortical segregation for other stimulus dimensions (Najafian et al., 2022). The question is whether this relation between retinotopy and ocular-dominance can be also demonstrated in mice.

There are different ways to address this question and the authors know best what approach is most appropriate. One possibility is to quantify retinotopy changes from the center of the ipsilateral cluster to the periphery as they did for ODI (e.g. Figure 1e-i). Another possibility is to compare the average retinotopy gradient between similar cortical sections in mice with and without ocular dominance clusters (e.g. cortical sections with the same average retinotopy in each animal). Another possibility is to compare the average retinotopy gradient between ocular dominance clusters of different sizes or different ODI indices. There can be several different outcomes from this effort: a) a relation between retinotopy and ocular dominance is found also in mice, b) there is no relation, c) the relation cannot be reliably measured. Whatever the outcome is, it would be helpful to report it. A quick comparison with the naked eye of Figure 1c and 1i panels appears to show an ipsi cluster centered in a cortical region dominated by elevation 24 deg, another one centered in a region dominated by elevation 0 deg and another centered in a region dominated by azimuth 24 deg (see screenshot below). However, proper quantification may tell a different story. Whatever the outcome is, it is worth reporting it.

We thank the reviewer for this suggestion, and we agree that both the presence, as well as the absence of inhomogeneities or gradients in the cortical retinotopic map aligning with the observed ocular dominance columns would be well worth investigating. Indeed, in our original approach we were not fully aware of the relevance of this recent work for our observations in mice. In fact, we had initially only acquired the retinotopic maps in order to make sure that our field-of-views were located in V1, not realizing that these maps would allow carrying out these analyses. We fully agree with the reviewer's point, and we have now added a substantial new section and an extra main figure to the manuscript, investigating the potential relationships between maps.

First, we created maps for ocular dominance, preferred azimuth and preferred elevation. For the ocular dominance maps, we applied an algorithm to precisely delineate regions that prefer the ipsilateral eye as well as regions preferring the contralateral eye. From the azimuth and elevation maps, we then calculated the cortical magnification factor (CMF; Daniel & Whitteridge, *J Physiol*, 1961), as well as the retinotopic distortion vector (Blasdel & Campbell, *J Neurosci*, 2001). In addition, we estimated the single neuron receptive field sizes from individual neuron tuning curves for azimuth position of visual stimuli.

Using these data, we investigated several relationships reported in earlier works, such as the angle between the retinotopic distortion vector and the orientation of the ODI gradient (Blasdel & Campbell, *J Neurosci*, 2001), the relation of ipsi-clusters to the cortical magnification factor and the receptive field size of neurons within ipsi-clusters (see lines 273-369, Fig. 3 and Extended data Fig. 10). These detailed analyses only indicated that ipsi-clusters were more likely to appear in the part of the retinotopic map that corresponds to the upper-frontal (i.e. binocular) visual field (Extended data Fig. 10d,e). Beyond this basic observation regarding the position of the clusters in the retinotopic map, we found no evidence for a significant relationship between cortical retinotopy and the ocular dominance map in mice.

In addition, the reviewer suggested to investigate whether mouse-to-mouse differences in cortical retinotopy correlated with the strength of ipsi-clusters. We investigated differences in "cluster quality" across animals using the following measures: (1) The overall number of detected ipsi-clusters, (2) the ODI-map value in the ipsi-cluster center (thus indicating how strongly the ipsi-cluster was dominated by the ipsilateral eye), and (3) "cluster quality" as per ranking by an experienced P.I. (who was blinded to other variables). We tested two hypotheses: 1) In mice with better cluster separation, visual channels for each eye would be separated more strongly, thus could require more cortical space for the duplicated retinotopic representations of each eye (Hubel & Wiesel, *Proc R Soc Lond B Biol Sci*, 1977; Tootell et al., *Science*, 1982; Blasdel & Campbell, *J Neurosci*, 2001; Najafian et al., *J Neurosci*, 2019) and 2) In order to support segregation of the visual channels for each eye, mice with better clusters could have a broader sampling of visual space by the thalamocortical projection at any given point in cortical space, which could translate into larger receptive fields in layer 4 (Najafian et al., *Nat Commun*, 2022). However, we did not observe significant relationships. Thus, mice with 'better clusters' do not appear to have a larger cortical magnification factor across binocular V1 (see "Reviewer Figure 3"), nor do they have overall larger receptive fields (see "Reviewer Figure 4").

The above-described novel analyses and interpretations have been added to the manuscript in the new section "*L4 ipsi-clusters show a non-random spatial arrangement, but no relationship with the retinotopic map*" on (lines 274-369), in Fig. 3, Extended data Fig. 10, and discussion (lines 435-446).

Reviewer Figure 3: Correlation of ‘ipsi-cluster quality’ with the cortical magnification factor.

Left: Cortical magnification factor (CMF) versus the ranking of an experienced P.I. (individual black dots represent mice, n=9). Middle: As ‘left’, but versus the number of ipsi-clusters in each mouse. Right: As ‘left’, but for the mean ODI in ipsi-clusters.

Reviewer Figure 4: Correlation of ‘ipsi-cluster quality’ with neuronal receptive field size.

Left: Receptive field (RF) size, in degrees azimuth, versus the ranking of an experienced P.I. (n=9 mice). Middle: As ‘left’, but for the number of ipsi-clusters in each mouse. Right: As ‘left’, but for the ODI in ipsi-clusters.

2) The data that the authors collected are very valuable to constrain future models of cortical topography. It would be helpful to provide detailed measurements of the ocular dominance segregation that they describe in the form of a table that includes animal number, number of neurons dominated by the contralateral and ipsilateral eye, number of neuronal clusters dominated by the ipsilateral eye, mean size of the ipsilateral clusters, and mean retinotopy change within and around the clusters. This information could be provided for each animal or only for animals that show the strongest ocular dominance segregation. There may be technical reasons that make this request not feasible, therefore, the authors can decide whether providing a table is a good idea or not.

We also see the benefit of providing the suggested table with the requested summary data per mouse within the paper and have added it to the manuscript (Table 1, lines 749-757).

In addition, we provide further and more detailed tabularized data in an excel file (Supplementary Data 1), which lists about 50 parameters describing properties and interactions of each ipsi-cluster, contralateral eye preferring region, the region between ipsi-clusters and contralateral eye dominated regions, and the imaged region corresponding to binocular V1.

Minor

1) Line 88. “irregularly features” should be “irregular features”.

We have corrected this.

2) Figure 1 legend. The scale bar is not reported for all panels (e.g. missing for panel c). If all scale bars are the same, it may be better to report the scale at the end of the figure legend (e.g. all scale bars are 100 microns).

We have updated all figure legends to include references to the scale bars.

3) “factors like... the retino-thalamo-cortical mapping ratio^{36–38} have been put forward in theoretical work to explain the absence of columns in a visual cortex as small as that of the mouse.”

The model from Najafian et al (reference 38) is not based on the retino-thalamo-cortical mapping ratio as references 36-37. It is based on the afferent density and size of the cortical area sampling the same retinotopy. Both Najafian et al. (2022) and Najafian et al. (2019) propose that ocular dominance columns emerge in the cortex of any species that has enough afferents (and cortical area) representing the same retinotopy (see screen shots from Najafian et al. 2019 representing the main idea below; the separation between the magenta lines represent cortical regions with non-overlapping receptive fields).

We apologize for this misrepresentation; it was the result of -very broadly- grouping multiple studies into a single short statement. We have corrected the statement, which reads now: “*Apart from taxonomy, factors like overall visual cortex size (Kaschube, 2014), the absolute number of neurons (Weigand et al., 2017), the retino-thalamo-cortical mapping ratio (Paik and Ringach, 2011; Jang et al., 2020) and the visual sampling density by geniculocortical afferents in visual cortex (Najafian et al., 2019, 2022) have all been put forward in theoretical work to explain the absence of columns in a visual cortex as small as that of the mouse.*” (lines 420-424).

4) “Possible explanations range from intracortical wirelength minimization^{27,35,45}, over merely being an epi-phenomenon created by the activity dependent wiring of cortical circuits⁴⁶, to a not very clearly spelled out function for binocular integration and stereoscopic depth perception⁴⁷.” This sentence should include the explanation that ocular dominance segregation emerges from an increase in the afferent density sampling each visual point, which enlarges the size of iso- retinotopic regions in visual cortex (Najafian et al., 2019).

The quoted sentence above refers to the (unanswered) question of how the presence of ocular dominance columns might improve visual processing. While we agree with the reviewer that the proposed addition gives an important explanation for how ocular dominance columns are formed, we believe that this explanation is better provided in the context of the discussion on mechanisms underlying the emergence of ocular dominance columns. Therefore, we have added a more detailed mechanistic explanation of how segregation of ocular dominance can emerge following the Najafian et al. (2019) model to the paragraph discussing implications for models in the discussion paragraph on lines 431-451.

References

- Blasdel, G., and Campbell, D. (2001). Functional retinotopy of monkey visual cortex. *J Neurosci* 21, 8286-8301.
- Hubel, D. H., and Wiesel, T. N. (1977). Ferrier lecture. Functional architecture of macaque monkey visual cortex. *Proc R Soc Lond B Biol Sci* 198, 1-59.
- Najafian, S., Jin, J., and Alonso, J. M. (2019). Diversity of Ocular Dominance Patterns in Visual Cortex Originates from Variations in Local Cortical Retinotopy. *J Neurosci* 39, 9145-9163.
- Najafian, S., Koch, E., Teh, K. L., Jin, J., Rahimi-Nasrabadi, H., Zaidi, Q., Kremkow, J., and Alonso, J. M. (2022). A theory of cortical map formation in the visual brain. *Nat Commun* 13, 2303.

Reviewer #3 (Remarks to the Author):

Goldstein and colleagues address the question to what extent mouse visual cortex exhibits a functional organization described in many other mammals, by focusing on eye preference and ocular dominance columns. Using large-scale calcium imaging in lightly anesthetized mice and drifting grating stimuli to the left or right eye, they identify ocular dominance columns in mouse primary visual cortex. These columns are characterized by an input preference from one eye over the other and stretch across cortical layers.

The study addresses an important problem in neuroscience, which is whether cortical columns are a general organizational principle of the mammalian cortex. The paper is well-written, and the data is presented clearly. The authors include many additional data and controls in the supplement and overall, the claims are supported by the data. However, I have two concerns:

First, in line 189 the authors write that “In several mice, pixelwise ODI maps across different cortical depths showed a clear similarity in the overall patterns of ipsilateral and contralateral eye dominated regions”. They should specify what “in several mice” means. Does that mean that the consistency across layers was only statistically significant for some mice? They discuss the variability across animals later, but it would be important to be more explicit in the Results section.

We thank the reviewer for the encouraging comments and questions. With “In several mice ...” we meant that in most mice we could clearly see a similar pattern. In addition, the quantitative comparison showed that the mean effect of lower ODI values in ipsi-clusters was significant when tested across the complete group of mice, and every single mouse showed lower ODI values across cortical layers for regions aligned with ipsi-clusters in layer 4 (compared to regions outside of the clusters, and local and global shuffle controls).

The consistence across individual mouse data can for instance be seen in Extended data Fig. 7c, grey lines. In this figure, the ODI for neurons in upper and lower L2/3, L4 and L5 is plotted relative to ipsi-clusters identified in layer 4. The individual mouse data (grey lines) show that some animals have more negative ODI values above/below L4 ipsi-clusters as compared to controls than other mice do, thus indicating variability. We did not test for significance of clustering effects in individual animals, but we explicitly direct the reader to these data when making statements on the variability across mice on lines 243-245: “*Similar to the within-L4 patterns of ipsi-clusters, the strength of the vertical alignment varied across mice (see Extended data Figs. 6, and 7c grey lines, for single mouse data across cortical layers).*”

In addition, some of the variability across mice is not mentioned and discussed at all. Fig. 1g shows that some mice have ipsi-clusters with a preference for the ipsilateral eye while the ipsi-clusters of other mice only have a weaker preference for the contralateral eye. This should be mentioned and discussed with respect to what is known about inter-animal variability with respect to ocular dominance columns. As much raw data for each animal should be shown as possible including in main figures.

Besides the broader mentioning of variability across mice as described above, we now explicitly refer to possible sources for variation in expression of ocular dominance columns in the discussion on lines 470-476: “*While we have not systematically explored the variability in the degree of columnar organization in our mouse data, there are clear and reproducible differences between individual animals (see e.g. maps in Extended data Fig. 10c). While this variability could originate from subtle differences in experimental conditions, it might as well result from natural variability in processes that give rise to ocular dominance columns and thereby provide a new opening for studying the role of columnar organization in supporting cortical computation.*”

In order to show as much individual mouse data as possible, we have now added a large figure panel containing, for each mouse, two separately imaged ODI maps, acquired with an interval of several

days to several weeks. The impact of variability across mice and variability due to methodological factors can be appreciated by comparing the differences between maps of individual mice, with the differences between maps of the same mouse imaged days or weeks apart (see Extended data Fig. 10c).

Second, the term “ocular dominance column” is not clearly defined in the paper. For example, would the existence of two ipsi-clusters be sufficient to call the observed functional organization a column? Or does it require a repetitive motif, as observed in other species with ocular dominance and orientation columns? The study still addresses an important question, but the observed organization looks rather like ocular dominance domains than columns. At the very least, the authors should define the term and discuss these problems.

We agree with the reviewer that the term “ocular dominance column” should be clearly defined in context of our finding. Throughout our manuscript, we use two types of terminology to describe the data, namely “clusters” and “columns”. With “clusters”, we refer to the increased densities of ipsilateral eye preferring neurons in layer 4, i.e. “ipsi-clusters”. The regions harboring these groups of neurons could in principle also be thought of as “patches” or “domains”, but because we detect them as groups of individual neurons with ipsilateral eye preference, we believe that “clusters” is the better term here as it directly describes the implemented method.

With “columns” we aim to describe the vertical alignment (across cortical layers) of ipsilateral eye preferring neurons, relative to the position of the layer 4 ipsi-clusters. As eye-preference is the functional property that aligns, we refer to this observation as columns for ocular dominance. Specifically, even if there would only be a few defined ipsi-clusters in layer 4, as long as these clusters extend vertically across the cortical layers, they would resemble groupings having the rough shape of columns, and therefore we would consider these ocular dominance columns.

Our definition of a column is now more explicitly referred to in the results section: “*Having found clusters of ipsilateral eye preferring neurons in layer 4 of mouse binocular V1, we asked whether these ipsi-clusters extended vertically into other cortical layers, i.e. show a columnar arrangement.*” (lines 195-196) and “*Thus, the ipsi-clusters we detected in cortical layer 4 extended vertically, in a columnar fashion, at least into cortical layers 2/3 and 5.*” (lines 245-246). Moreover, we now also introduce this very definition in the abstract of the manuscript “*Here, we report the discovery of clusters of ipsilateral eye preferring neurons in layer 4 of the mouse primary visual cortex. These clusters extend into layer 2/3 and upper layer 5, forming a columnar pattern for ocular dominance.*” (lines 29-31).

Finally, in the discussion we added a new section addressing the question of whether or not the ocular dominance columns, as we observe in mice, can be directly compared to other species, or whether they might be a fundamentally different form of cortical micro-organization (lines 397-418).

REVIEWER COMMENTS

Reviewer #1 (Remarks to the Author):

My initial concern during the 1st round was that the results and analyses presented were insufficient to support the claim that the cluster of ipsi-preferring neurons constitutes a columnar structure. The authors have sincerely addressed this concern by providing additional analyses and discussions that have led to a more accurate understanding of the ocular dominance (OD) structure in mice. In higher mammals, OD columns consist of completely segregated groups of cells that respond to either ipsi or contra inputs, with these ipsi- or contra-preferring groups of cells extending across cortical layers. However, the OD structure discussed in this paper, specific to mice, differs from those in higher mammals. In mice, clusters of cells that respond more strongly to ipsilateral inputs exist at some locations within the binocular region of V1 and are mixed with contra-preferring neurons and binocular neurons. So I still have concerns about the terminology about the “columnar” organization of ocular dominance. In addition, there are some unclear points in the additional analyses. To make this paper more accurate, I believe the authors need to revisit these points once again.

We thank the reviewer for the further comments and suggestions, which we find generally very helpful. Below, we specify point-by-point (in blue text) how we have implemented each in our manuscript.

Major comments

1) My main concern is with the terminology related to “columns”. In the revised manuscript, the authors define the vertically extending clusters as columns. However, this term naturally reminds many readers of functional clusters that extend consistently in shape across layers, as observed in cats and monkeys. The data shown in the manuscript (e.g., Extended data Figures 6, 7, 9) indicate that the cluster shapes are only weakly similar across planes. For example, in the new result shown in Extended data Figure 9, although the peaks are significantly different from the shuffled control, the differences are very small. This suggests that the cross-correlation peaks are mostly due to the global structure, while the contributions of the local structures are very small. Further, as seen in Extended data Figure 6, the ODI map patterns are only weakly preserved between L4 and other layers. This type of functional organization is far from what the OD columns usually observed in other species. Therefore, I recommend that the authors use other terms and accurately describe the results without exaggeration throughout the manuscript including the title and abstract.

Regarding this first point about the terminology, the reviewer indicates in the general assessment above that ocular dominance columns in the visual cortex of higher mammals are “completely segregated groups of cells that respond to either ipsi or contra inputs, with these ipsi- or contra-preferring groups of cells extending across cortical layers”. While we understand the principal point the reviewer is making, we would like to point out that the more-or-less complete segregation really only applies to -monkey- primary visual cortex layer IVc. Neurons in the more superficial as well as the deeper cortical layers show integration of responses from both eyes, that is, these neurons are to some degree binocular, though typically preferring one eye over the other eye. This can, for instance, be appreciated in Figure 13 of Hubel and Wiesel’s Ferrier lecture (Proceedings of the Royal Society B: Biological Sciences, 1977). This figure shows that most neurons in layer II-III are of OD groups 2-3 and 5-6, meaning that they are being driven by both eyes. Hubel and Wiesel further indicate that this observation extends to nearly all layers of cortex: “In layers II and III (and also in V and VI) most cells are binocular but show some eye preference.” (see legend of Figure 13 and also the schematic in Figure 11). In contrast, in layer IVc most neurons indeed are in OD groups 1 and 7, indicating that these are driven by only one eye (Figure 13, right panel). Thus, while the segregation of neurons by eye preference in monkey (and cat) visual cortex is certainly more defined in comparison to what we found in the mouse, the suggested complete

segregation of eye responses throughout cortical layers is not a phenomenon generally observed in higher mammals.

Notwithstanding, we see the reviewer's point that we should avoid the impression that the functional organization we observe in mice is a one-to-one copy of what has been found in monkey and cat visual cortex (see e.g. also the Discussion section of our manuscript, second paragraph). In order to make this even clearer, we changed how we refer to the vertically extending structures and named them now 'column-like structures'. This term indicates that there is a vertical extent to the functional organization we observe, but explicitly does not use the designation 'column' as is. The term column-like reflects that the functional organization for OD looks to some degree like a column, but that at this point it cannot be determined whether these are in actuality the same cortical columns as have been defined for higher mammals.

2) Regarding the cross-correlation map in Figure 2f, the peaks in the cross-correlation maps are still unclear. Extended data Figure 9 is important to show quantitatively that the peaks of the cross-correlation maps had very small differences from the shuffled control and should be presented in a main figure. In the statistical analyses of Extended data Figure 9, the Kruskal-Wallis test gives a very small p-value of 10^{-36} , even though there is little difference from the shuffled control, suggesting that perhaps the authors may be using the degrees of freedom incorrectly. I speculate that the Wilcoxon matched-pairs signed rank test was applied at each distance point and repeated many times and the multiple comparison problem is not properly addressed.

In order to make the peaks in the cross-correlation maps clearer, we have added insets with individually adjusted scaling (Figure 2 and Extended data Figure 8). In these insets, the peaks are clearly visible.

Next, by the reviewer's request we have moved the panels from Extended data Figure 9 to main Figure 2. We believe that this helps the reader appreciate that even 'mild' shuffling of data (i.e. swapping ODI values within a range of 200 μm) already significantly reduces the peak of the cross-correlation maps. In addition, these panels of previous Extended data Figure 9 also show the effect of shuffling ODI values altogether, which completely removes the peak in the cross-correlation maps. Together, we believe that these data do not merely show that the effect of 'mild' shuffling is small, but rather that beyond the broader ODI map patterns, there is a fine-grained pattern in the ODI map that significantly replicates across cortical layers.

Finally, we thank the reviewer for pointing out a potential issue with the Kruskal-Wallis test we had used. We have reconsidered the problem and have realized that within the cross-correlation maps there might be correlations among nearby datapoints (e.g. as a result of the sliding window approach and data smoothing), which affects the independence of the individual datapoints across the spatial axis of the plot. In order to address this problem, we now use a linear mixed-effects model (LMM) to test for a difference between the two conditions (real data vs 200 μm swapped data) and the interaction of this effect with the spatially repeated measurements (position along the x-axis or y-axis of the correlation map). The interaction term in the model takes into account the possibility of spatial correlations, and from this term we can read out at which positions along the spatial axis the two conditions are significantly different. The analysis showed that the regions around the peaks are significantly different from the shuffle control (200 μm swapped data). We have updated the reporting of these statistical analyses in the legend of Figure 2 and added its description in the statistics subsection of the Methods section of the manuscript (on lines 748-754).

3) The authors have demonstrated that neurons responding to ipsilateral inputs form clusters. However, it remains unclear whether these neurons are ipsi-preferring neurons or binocularly responsive neurons. As the authors noted, the average ODI (Ocular Dominance Index) is close to zero even near the center of the clusters (Fig. 1g). Based on the current data, it is unclear whether this is due to the mixing of ipsi- and contra-preferring cells or simply because the clusters are composed of binocularly responsive neurons. To clarify this, the authors should present the ODI distribution of individual neurons from the ipsi-preferring clusters. If there are many cells with ODIs below some threshold (e.g. -0.3) in the clusters, they would be considered ipsi-preferring clusters. On the other hand, if neurons with ODIs close to zero predominate, they would be considered binocularly responsive clusters. If the latter is the case, I believe that the paper would be more accurate if it discussed binocularly responsive clusters rather than ipsi-preferred clusters. In the additional analysis to determine the percentage of ipsi-preferring neurons (Extended Data Figure 4), the authors cut off the ipsi-preferring neurons at $ODI = 0$, and if they are even slightly negative, they call them ipsi-preferring neurons. However, for example, neurons with $ODI = -0.01$ cannot be called ipsi-preferring neurons, but rather binocular neurons. The authors should define ipsi-preferring neurons when the response of ipsi is statistically significantly greater than that of contra, or when some threshold (e.g. -0.3) is set on the ODI.

To investigate the distribution of ODI values in density-based ipsi-clusters, we plotted histograms of ODI distributions of cells at different distance ranges from the centers of ipsi-clusters (new Extended data Figure 4d). These data show that at the center of ipsi-clusters, the ODI distribution is skewed towards negative values, indicating ipsi-eye dominance. Accordingly, we believe the term ipsi-clusters is justified, and we would like to keep it.

We also followed the reviewer's suggestion and tested whether changing the ODI threshold or using the criterion of a significant difference between ipsi- and contra responses alter our results in Extended data Figure 4a-c. We have analyzed the data with both of these suggested definitions of ipsilateral eye preferring neurons, and the results essentially show the same pattern in relation to ipsi-clusters. The only difference is an overall offset of the percentage of ipsilateral eye preferring neurons. We have added a statement reflecting these findings to the Results section (on lines 151-154) describing the fraction of ipsilateral eye preferring neurons: "*The fraction of ipsilateral eye preferring neurons ($ODI < 0$) was larger near ipsi-clusters (Extended data Fig. 4a-c; this also held for ipsilateral eye preferring neurons with $ODI < -0.3$, or for neurons responding significantly stronger to the ipsilateral compared to the contralateral eye; data not shown).*". In addition, we show the data here for the reviewer in Reviewer Figure 1, below.

Reviewer Figure 1. Fraction of ipsi-prefering neurons relative to ipsi-cluster centers, calculated using different definitions of ‘ipsi-prefering’

a-c, Fraction of ipsi-prefering neurons as a function of distance to ipsi-clusters; same data as Extended data Figure 4a-c. **a**, Comparison of ODI “In” (100 μm range) and “Out” (100 μm -200 μm) of ipsi-cluster centers. **b**, Comparison of original data with global randomization controls, shuffled ODI values (blue), random XY positions (green) and randomly sampled cluster positions. **c**, As **b**, but for local randomization control (swapping ODIs of pairs of neurons). **d-f**, As **a-c**, but for ipsi-prefering neurons defined as having an ODI smaller than -0.3. **g-i**, As **a-c**, but for ipsi-prefering neurons defined as having a significantly stronger response to the ipsilateral eye as compared to the contralateral eye. All panels: ns: not significant, * $p < 0.05$, ** $p < 0.01$, $n = 9$ mice.

Minor points

L653-655: According to L653, 1-5 clusters were detected in each volume. On the other hand, according to L655, the top three clusters were used in the “group-wise quantitative analyses”. Does this mean that in each mouse, multiple volumes were recorded, at least three clusters were detected, and the top three clusters among them were averaged in each mouse and the averaged values were used for the “group-wise quantitative analyses” with a sample number of 9mice? How many clusters were detected by the method (ipsi cluster detection in L633-655) in each mouse? Describe these points clearly. Also specify in which analyses or figures “the group-wise analyses” are used.

We thank the reviewer for pointing out that this was not clear. In each mouse ($n=9$), we recorded one L4 volume. In that volume, we detected the clusters using the density clustering based approach. In the majority of mice, we detected 3 clusters ($n=6$) and in the other we detected 1,2 and 5 clusters. However, in some randomization control data sets, the clustering algorithm detected more clusters. In order to compare between control and real data, we therefore included only the best three clusters (i.e. having the highest ρ_i and δ_i values) if the clustering algorithm detected more than three clusters. This way, we ensured that the control datasets did not have ‘poorer clusters’ merely because we included more low quality clusters.

We have added a paragraph in the Methods section (on lines 637-646) that details this approach better, and reports on the numbers of clusters observed, and used in analyses: “Typically, in the L4 volumes (one volume per mouse, spanning about 50 percent of their binocular visual cortex area), the algorithm detected one to five ipsi-cluster centers (mean: 2.9 ± 1.0 s.d.; M01, M02, M03, M05, M06 and M08: 3 clusters; M04: 1 cluster; M07: 5 clusters; M09: 2 clusters). However, in some randomized control datasets (see below), the algorithm detected on average more clusters (Shuffle ODI: 3.9 ± 1.3 s.d.; Uniform XY: 5.2 ± 1.2 s.d.; Random clusters: 2.9 ± 1.0 s.d.). In order to increase comparability between the real data and the control datasets, we included only the three clusters with highest ρ_i and δ_i values in group-wise quantitative analyses. The mean numbers of clusters per L4 volume (i.e. per mouse) in those analyses were therefore for real data $2.7 (\pm 1.0$ s.d.), for shuffle ODI data $2.9 (\pm 0.4$ s.d.), for uniform XY data $3.0 (\pm 0.1$ s.d.) and for random clusters data $2.7 (\pm 0.7$ s.d.).”

A new method was used to geometrically identify ipsi- and contra-clusters. According to the L654-655 and the Table 1, the number of clusters is different between those included in the “group-wise quantitative analyses (L654)” and those geometrically detected by the new method (Table 1). How many clusters overlap between them?

As explained in the revised manuscript, the density-based clustering method did not provide us with an outlined area for the ipsi-clusters, but only with the cluster centers. Thus, we cannot calculate a measure of overlap directly. However, we did investigate whether the density-based cluster centers fell within the geometry-based cluster outlines, which turned out to be the case for most density-based ipsi-clusters.

We now provide these data in the Methods section (on lines 725-735) of the revised manuscript: “We verified that the two methods for detecting ipsi-clusters resulted in similar outcomes by quantifying the number of density-based ipsi-clusters that fell within the outlines of geometrically defined ipsi-clusters. Note that we could not directly calculate the overlap of clusters detected by the two different methods because the density-based method only provided center coordinates for ipsi-clusters. From the 26 density-based ipsi-clusters, 18 fell within the outline of a geometrically defined ipsi-cluster, and three were located right outside of the edge of a geometrically defined ipsi-cluster. Two clusters fell outside the region defined as binocular VI, and three density-based ipsi-clusters were located in regions that exhibited low ODI values, but were not part of geometrically defined ipsi-clusters. Thus, the vast majority of density-based ipsi-clusters were detected in regions that were part of geometrically defined ipsi-clusters.”

In addition, we provide a visualization of these data for the reviewer, see below (Reviewer Figure 2).

Reviewer Figure 2. Comparison of density-based ipsi-clusters and geometry-defined ipsi-clusters. V1b: Largest black outlined region. Geometry-identified ipsi-clusters: Smaller black outlined regions. Density-based ipsi-clusters: Black-white crosses.

The explanations of the new method in the Methods (L721-733) seem to be inconsistent with those in the legend of Extended data Figure 10 (L959-977). For example, the thresholds for the high-pass filtered maps were different (15th percentile in L724 vs. 25th percentile in L962). Please describe the explanation consistently in these two parts.

We thank the reviewer for spotting the mistake in Extended data Figure 10's legend (the explanation in the Methods section of the manuscript is the correct one). In short, the threshold for the high-pass filtered maps was the 15th percentile. However, an additional criterium for the geometry-defined ipsi-clusters was that the actual (not highpass) ODI value at the center coordinate was lower than the 25th percentile of actual ODI map values in the V1b region. We have corrected this in the legend of Extended data Figure 9 of the revised manuscript.

L722: In “ by subtracting two ODI maps”, please specify which map was subtracted from which map (as in L962).

We have added this information to the Methods section (on lines 711-714): “*Geometrically defined ODI clusters were identified in a highpass-filtered ODI map, which was calculated by subtracting two ODI maps, smoothed with kernels that differed in size by a factor of 2.5 (i.e. subtracting a map smoothed with $\sigma=42 \mu\text{m}$ from a map smoothed with $\sigma=105 \mu\text{m}$; see Extended data Fig. 9a).*”

L263: Please add explanation of term, “V1b” on first use.

We have added the definition of V1b (binocular region of the primary visual cortex) on line 267.

L288: typo. 'In order to asses...' should be corrected to 'In order to assess...!'

Thank you, the typo has been corrected (line 289).

In the middle and bottom panels of Extended data Figure 7a and b, please indicate the reference (or origin) points to which the ipsi-cluster centers are aligned. In addition, it should be helpful to draw rough boundaries between layers.

The ipsi-clusters were always aligned to the central axis of the side-views, which is indicated with a black vertical line. We have added this explanation to the legend of Extended data Figure 7. In addition, we have added demarcations of the depth from the surface and approximate cortical layer in Extended data Figure 7.

In the Scnn1a-cre mice, data were obtained from the right hemisphere. From which hemisphere were the other data recorded? The horizontal flip of the maps between Extended data Figure 5c and other related maps may confuse the readers unfamiliar with the mouse visual cortex.

We have added the position of the cranial window for both GCaMP6s transgenic mice and Scnn1a-Tg3-Cre mice in the methods section (on lines 492-493): “*The cranial windows were placed over the visual cortex of the left hemisphere in GCaMP6s transgenic mice and the right hemisphere in Scnn1a-Tg3-Cre mice.*”

In addition, we have added a note to the legend of Extended data Figure 5 specifying the horizontal flip of the map as a result of using data from the other hemisphere.

Reviewer #2 (Remarks to the Author):

The authors have fully addressed all my comments. Their careful analysis and detailed functional measurements of mouse visual cortex (including the extensive supplementary data) provide new important constraints to developmental models of visual cortical topography.

We thank the reviewer for the feedback and the helpful comments in the first round of revisions.

This paper demonstrates subtle but significant neuronal clustering for ocular dominance in the mouse visual cortex. This finding is important because the mouse has become a popular model to study visual cortical circuitry and all studies are currently designed under the assumption that neurons dominated by the ipsilateral eye are randomly interspersed with the more common neurons dominated by the contralateral eye. Ocular dominance columns are thought to be only present in animals with larger binocular fields such as carnivores and primates and some types of rats, but not mice. The finding that small ocular dominance domains can be found in the tiny binocular cortex of a mouse provides new important constraints to developmental models of cortical topography.

The paper is well written, the main results are well documented, the conclusions are solid, and the authors have performed several important controls to clearly demonstrate that the minute ocular dominance clusters are not the result of random (salt-and-pepper) cortical topography. My main comment is that the authors do not seem to be aware of recent work proposing a mechanism of ocular dominance segregation based on the density of thalamic afferents per visual point. According to this mechanism, ocular dominance columns emerge when the number of afferents with overlapping receptive fields (and the iso-retinotopic cortical area accommodating the afferents) are large enough to allow the segregation of afferents and cortical neurons by eye input. Importantly, this mechanism predicts a close relationship between ocular dominance segregation and the retinotopy gradient, which has been demonstrated in humans, macaques and cats (Blasdel and Campbell, 2001; Najafian et al., 2019). The question is whether such relationship also exists in mice. The data from the authors may allow to address this question and it would be a missed opportunity not to do it by performing some additional analysis that should not be too time consuming. More details below.

Major comments

1) Ocular dominance segregation is closely related to cortical retinotopy in several species of mammals. Evidence supporting this relation can be traced back to the work of Hubel and Wiesel demonstrating that ocular dominance stripes run orthogonal to the cortical border between areas V1 and V2 in macaques (Hubel and Wiesel, 1977) and the work of Campbell and Blasdel demonstrating that cortical retinotopy changes slower along than across the V1/V2 border (Blasdel and Campbell, 2001). More recently, a relation between the retinotopic gradient and ocular dominance segregation has been demonstrated for a large portion of area V1 in humans, macaques and cats (Najafian et al., 2019). This close relation between retinotopy and ocular dominance segregation supports the notion that thalamic afferents segregate by eye input in visual cortex when the cortical area representing the same retinotopy is large enough to accommodate neuronal clusters with different eye dominance (Najafian et al., 2019). The same mechanism has been also proposed to explain cortical segregation for other stimulus dimensions (Najafian et al., 2022). The question is whether this relation between retinotopy and ocular-dominance can be also demonstrated in mice.

There are different ways to address this question and the authors know best what approach is most appropriate. One possibility is to quantify retinotopy changes from the center of the ipsilateral cluster to the periphery as they did for ODI (e.g. Figure 1e-i). Another possibility is to compare the average retinotopy gradient between similar cortical sections in mice with and without ocular dominance clusters (e.g. cortical sections with the same average retinotopy in

each animal). Another possibility is to compare the average retinotopy gradient between ocular dominance clusters of different sizes or different ODI indices. There can be several different outcomes from this effort: a) a relation between retinotopy and ocular dominance is found also in mice, b) there is no relation, c) the relation cannot be reliably measured. Whatever the outcome is, it would be helpful to report it. A quick comparison with the naked eye of Figure 1c and 1i panels appears to show an ipsi cluster centered in a cortical region dominated by elevation 24 deg, another one centered in a region dominated by elevation 0 deg and another centered in a region dominated by azimuth 24 deg (see screenshot below). However, proper quantification may tell a different story. Whatever the outcome is, it is worth reporting it.

2) The data that the authors collected are very valuable to constrain future models of cortical topography. It would be helpful to provide detailed measurements of the ocular dominance segregation that they describe in the form of a table that includes animal number, number of neurons dominated by the contralateral and ipsilateral eye, number of neuronal clusters dominated by the ipsilateral eye, mean size of the ipsilateral clusters, and mean retinotopy change within and around the clusters. This information could be provided for each animal or only for animals that show the strongest ocular dominance segregation. There may be technical reasons that make this request not feasible, therefore, the authors can decide whether providing a table is a good idea or not.

Minor

1) Line 88. “*irregularly features*” should be “*irregular features*”.

2) Figure 1 legend. The scale bar is not reported for all panels (e.g. missing for panel c). If all scale bars are the same, it may be better to report the scale at the end of the figure legend (e.g. all scale bars are 100 microns).

3) “*factors like... the retino-thalamo-cortical mapping ratio^{36–38} have been put forward in theoretical work to explain the absence of columns in a visual cortex as small as that of the mouse.*”

The model from Najafian et al (reference 38) is not based on the retino-thalamo-cortical mapping ratio as references 36-37. It is based on the afferent density and size of the cortical area sampling the same retinotopy. Both Najafian et al. (2022) and Najafian et al. (2019) propose that ocular dominance columns emerge in the cortex of any species that has enough

afferents (and cortical area) representing the same retinotopy (see screen shots from Najafian et al. 2019 representing the main idea below; the separation between the magenta lines represent cortical regions with non-overlapping receptive fields).

[REDACTED]

4) *“Possible explanations range from intracortical wirelength minimization^{27,35,45}, over merely being an epi-phenomenon created by the activity dependent wiring of cortical circuits⁴⁶, to a not very clearly spelled out function for binocular integration and stereoscopic depth perception⁴⁷.”* This sentence should include the explanation that ocular dominance segregation emerges from an increase in the afferent density sampling each visual point, which enlarges the size of iso-retinotopic regions in visual cortex (Najafian et al., 2019).

References

- Blasdel, G., and Campbell, D. (2001). Functional retinotopy of monkey visual cortex. *J Neurosci* 21, 8286-8301.
- Hubel, D. H., and Wiesel, T. N. (1977). Ferrier lecture. Functional architecture of macaque monkey visual cortex. *Proc R Soc Lond B Biol Sci* 198, 1-59.
- Najafian, S., Jin, J., and Alonso, J. M. (2019). Diversity of Ocular Dominance Patterns in Visual Cortex Originates from Variations in Local Cortical Retinotopy. *J Neurosci* 39, 9145-9163.
- Najafian, S., Koch, E., Teh, K. L., Jin, J., Rahimi-Nasrabadi, H., Zaidi, Q., Kremkow, J., and Alonso, J. M. (2022). A theory of cortical map formation in the visual brain. *Nat Commun* 13, 2303.